# Decoding Large Language Diffusion Models with Foreseeing Movement

## Abstract

Large Language Diffusion Models (LLDMs) benefit from a flexible decoding mechanism that enables parallelized inference and controllable generations over autoregressive models. Yet such flexibility introduces a critical challenge: inference performance becomes highly sensitive to the decoding order of tokens. Existing heuristic methods, however, focus mainly on local effects while overlooking long-term impacts. To address this limitation, we propose the Foreseeing Decoding Method (FDM), a novel approach that integrates both local and global considerations to unlock the full potential, employing a search-based strategy to enable effective optimization in discrete spaces. Furthermore, by analyzing the consistency of chosen tokens in the full decoding process, we develop a variant, FDM with Acceleration (FDM-A), which restricts deep exploration to critical steps identified as the exploration and balance circumantences. Extensive experiments across diverse benchmarks and model architectures validate the scalability of FDM and demonstrate the superior efficiency-performance trade-off achieved by FDM-A. Our work might potentially provide a principled step toward more powerful decoding methods for LLDMs.

## 1 Introduction

In recent years, diffusion models (Ho et al., 2020; Rombach et al., 2022a) have emerged as a promising competitor in natural language modeling (Nie et al., 2025; DeepMind, 2025), challenging the dominance of auto-regressive generation (Jaech et al., 2024; Guo et al., 2025; Team et al., 2025). Instead of generating tokens sequentially from left to right and token by token, Large Language Diffusion Models (LLDMs) can generate multiple tokens in parallel,

Table 1: The performances of LLaDA with various decoding orders on the ARC (Clark et al., 2018) benchmark.

|  | Random | Margin | FDM-A |
|---|---|---|---|
| Accuracy (↑) | 79.06 | 82.55 | **86.30** |
| Tokens/Seconds (↑) | 12.01 | 10.85 | **38.20** |

making it highly efficient at the inference stage (Li et al., 2025; Khanna et al., 2025). In addition, owing to pipelines of processing denoising and the bidirectional modeling, LLDMs can outperform the auto-regressive models in broad aspects such as reverse reasoning (Berglund et al., 2024), controllable generation (Xiong et al., 2025; Li et al., 2022), or multi-modal reasoning (Li et al., 2025).

However, high flexibility is a double-edged sword, as it can lead to performance degradation if sampling paths are chosen inappropriately, attributing to the convergence issues (Li & Cai, 2025) and the non-convexity of the optimization goal (Kim et al., 2025; Peng et al., 2025; Ben-Hamu et al., 2025). For example, as shown in Table 1, compared to decoding with a random order, decoding answers with the order of the largest marginal probability yields an improvement on the ARC benchmark (Clark et al., 2018) from 79.06% to 82.55%, underscoring its critical importance. To address this issue, previous works have mainly developed strategies from a heuristic perspective. For example, pioneer works (Nie et al., 2025; Zheng et al., 2024) propose to decode by enlarging the predicted probability. Follow-up works argue that probability margins (Kim et al., 2025) and the entropy of the predicted distribution (Ben-Hamu et al., 2025) are better substitutes, considering that LLDMs may be confused when the probabilities of the topmost tokens at the given position are near. However, though effective, previous heuristic approaches decode tokens with local information at each step, overlooking the long-term impacts underlying the full decoding paths without leveraging the full potentiality of LLDMs at the inference stage.

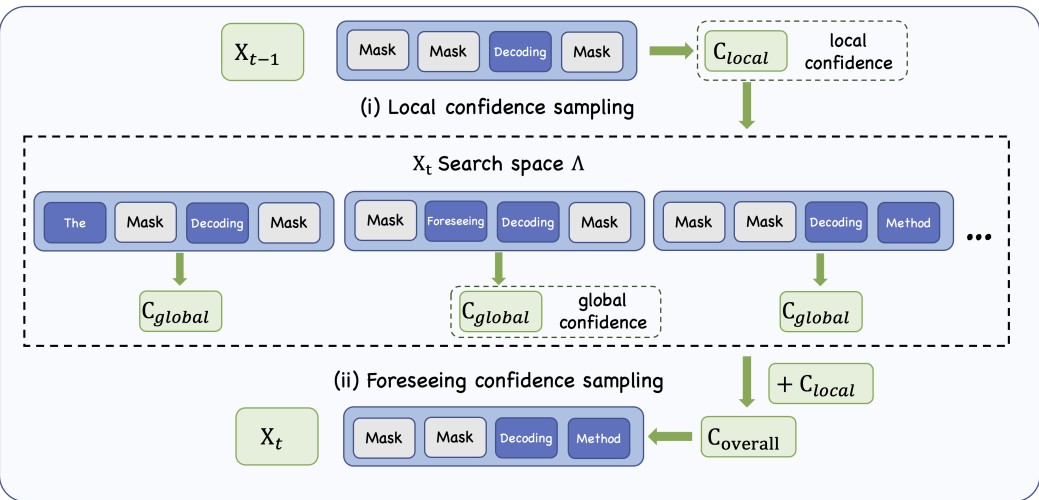

Figure 1: The pipeline of FDM. We first compress the search space into a small set $\Lambda$ by filtering out candidates of lower local confidence *i.e.* $C_{local}$. In the final, we incorporate both local and global confidence to decide the ultimate choice at step $t$.

Therefore, in this paper, with further analyzing the decoding formulation of LLDMs, we find that it can be divided into two components. The first one is termed as **local confidence** in Figure 1 (i), reflecting the uncertainty of models in predicting a given token and widely adopted by the heuristic methods. The other is the **global confidence**, capturing the future impact in decoding a specific token. It is not directly accessible and typically ignored by prior methods. However, we demonstrate that the training goal of LLDMs provides a good approximation. By further combining both of them into considerations, we propose **F**oreseeing **D**ecoding **M**ethod (FDM), illustrated in Figure 1. To ensure the computation is affordable in reality, the decoding process of FDM is divided into two stages: We firstly rank the candidate tokens with their local confidence. In the second competition, the overall criterion of both local and global confidence is adopted to decide the final tokens chosen for decoding. We not only theoretically prove FDM will achieve lower errors compared to heuristic methods but also perform experiments across multiple benchmarks to illustrate that the benefit of it can further improve by enlarging the search space. It demonstrates FDM will serve as a test-time scaling (Snell et al., 2024; Bi et al., 2024) method specifically designed for LLDMs.

Furthermore, by calculating the consistency ratio of FDM and the simple confidence-based decoding, we find that a global view is not always necessary at every step, especially when the context is sufficient for reliable decisions. Motivated by this observation, we propose an accelerated version of FDM (FDM-A). It will perform an adaptive exploration from the probability feedback, switching to FDM only when all candidates have low confidence or borderline scenarios. As shown in Table 1, FDM-A can not only accelerate the decoding process with over $3\times$ speed-up but also obtain notable performances. In summary, our contributions are listed as follows:

- We propose the **F**oreseeing **D**ecoding **M**ethod (FDM) to schedule the decoding orders of LLDMs. Unlike heuristic approaches, it takes both the local and global confidence as the criterion for decision, guaranteeing a lower divergence with the natural distribution.

- Motivated by the agreement ratio in decoding, we further accelerate FDM with an adaptive strategy (FDM-A). It adaptively performs exploration when the context is rare and parallel decodes tokens if models have enough context for complete generation.

- The experiments across multiple benchmarks and variants of LLDMs are performed, manifesting the scalability of FDM and the outstanding performance of FDM-A in balancing efficiency and performance.

## 2 RELATED WORK

**Large Language Diffusion Models (LLDMs).** As one of the most successful types of generative models, the popularity of diffusion models has been well-recognized in the vision domain (Ho et al., 2020; Peebles & Xie, 2023; Rombach et al., 2022b). At the training stage, the model predicts the noises added to the natural inputs. Then, at the test time, by utilizing the model to denoise step by step, diffusion models recover the original data through iterative refinement. Inspired by their great success, early investigations such as D3PMs (Austin et al., 2021), DiffuSeq (Gong et al., 2023) have demonstrated the potential of the diffusion pipeline in language tasks. However, limited to the small scales, their performances are considerably inferior to those of their auto-regressive competitors. Recently, LLaDA (Nie et al., 2025; Zhu et al., 2025) scales the model parameters to billions, considered as one of the most representative works in Large Language Diffusion Models (LLDMs). To save the computational costs, other architectures like Dream (Ye et al., 2025), DiffuGPT (Gong et al., 2024), and Dream-Coder (Xie et al., 2025) perform continual training on the pretrained weights of the auto-regressive counterparts. Built upon the outstanding performances in the language capacity, large language diffusion models have also been applied to multi-modal applications such as biomedical understanding (Dong et al., 2025), chart understanding (You et al., 2025), and mathematical reasoning (Yang et al., 2025). But in this paper, we propose FDM to improve the performance of LLDMs at the inference stage.

**Decoding Order of LLDMs.** With the rapid development of LLDMs, their shortcomings are also disclosed. One of the most prominent ones is the vital significance of the decoding orders (Kim et al., 2025). This is because when the context is limited, models will fail to accurately predict all tokens. In other words, the uncertainty of predictions can be measured with the output probability. This motivates the heuristic-based approaches, including decoding with the largest probability value (Nie et al., 2025; Zheng et al., 2024), with the largest margin probability (Kim et al., 2025), and with the least entropy of distribution (Ben-Hamu et al., 2025). Although heuristic methods improve the performance of LLDMs a lot compared to decoding with the random order, they make decisions based on local confidence, ignoring the sequential consequence to other tokens. This limitation may bring errors to the generated sequences. Thus, in this paper, we propose FDM and FDM-A, which incorporate the global confidence to depict the future cascading effect in the decoding process. .

**Acceleration of LLDMs.** Although LLDMs map to the complete answer at each step, they will suffer a quality-efficiency trade-off with a varying decoding step (Feng et al., 2025; Nie et al., 2025). In addition, to achieve the bidirectional attentions, the attention maps of LLDMs are not causal, requiring model-agnostic methods like KV-Cache for adaptations. Therefore, in a series of works (Wu et al., 2025; Hu et al., 2025; Ma et al., 2025), they propose their own approximation acceleration for reusing the Key-Value, largely accelerating the inference speed. More related to our work, Hong et al. (2025) and Ben-Hamu et al. (2025) both propose samplers for dynamic decoding in LLDMs. But Hong et al. (2025) purely focus on the parallel decoding for acceleration and Ben-Hamu et al. (2025) propose WINO for revoking suspicious error tokens. While in our paper, we propose FDM-A to balance the explorations and accelerations in decoding, achieving a better trade-off.

## 3 PRELIMINARIES

Unlike auto-regressive models that generate answers in a sequential way, LLDMs define a Markov chain that revises the answer step-by-step with a denoised process. Given the user query $\mathbf{q}$ and the vocabulary space $M = \{1, 2, ...m\}$, the generation of final answer $\mathbf{x}_T$ with length $L$ under the data distribution $p_{data}$ can be formulated as:

$$\mathbf{x}_T = \arg\max_{\mathbf{x}_T} p_{data}(\mathbf{x}_{0:T}|\mathbf{q}) = \arg\max_{\mathbf{x}_T} p(\mathbf{x}_0) \prod_{\alpha=1}^{T-1} p_{data}(\mathbf{x}_\alpha|\mathbf{q}, \mathbf{x}_{0:\alpha-1}), \quad (1)$$

where $p(\mathbf{x}_0)$ is sampled from an initial noise distribution which is independent to $\mathbf{q}$. $\mathbf{x}_\alpha$ is a partially masked sequence at the $\alpha$ step. To analyze the influence of the intermediate variable $\mathbf{x}_t$, we can regroup the terms associated with step $t$ as:

$$\mathbf{x}_T = \arg\max_{\mathbf{x}_T} p_{data}(\mathbf{x}_{t+1:T}|\mathbf{q}, \mathbf{x}_{0:t}) p_{data}(\mathbf{x}_t|\mathbf{q}, \mathbf{x}_{t-1}) \prod_{\alpha=1}^{t-1} p_{data}(\mathbf{x}_\alpha|\mathbf{q}, \mathbf{x}_{0:\alpha-1}). \quad (2)$$

Although the above equation shows that $\mathbf{x}_t$ can impact future variables from $\mathbf{x}_{t+1}$ to $\mathbf{x}_T$, heuristic decoding methods commonly simplify the procedure by focusing solely with the local partition:

$$\pi_H(\mathbf{x}_t|\mathbf{x}_{t-1}) : \mathbf{x}_t = \arg\max_{\mathbf{x}_t} p_\theta(\mathbf{x}_t|\mathbf{q}, \mathbf{x}_{t-1}), \tag{3}$$

where $\theta$ is a parameterized neural network introduced by LLDMs to model the natural distribution. According to (Nie et al., 2025), the optimization target of LLDMs can be given as:

$$\mathbb{E}_{\mathbf{x}_T \sim p_{data}, t \in [0,T]} \frac{1}{n} \sum_{j=1}^{n} \mathbf{1}[\mathbf{x}_t^{(j)} = \texttt{Mask}] \odot \log p_\theta(\mathbf{q}, \mathbf{x}_t^{(j)})[\mathbf{x}_T], \tag{4}$$

where $\odot$ is the Hadamard product and $[\mathbf{x}_T]$ represents the probability value of the tokens in $\mathbf{x}_T$. With a well-trained model, LLDMs learn to approximate the conditional log-probability of future tokens in $\mathbf{x}_m$ given the current masked state, $\mathbf{x}_t$ ($m > t$) and the user query $\mathbf{q}$. Specifically,

$$\log p_\theta(\mathbf{x}_m|\mathbf{x}_t, \mathbf{q}) = \mathbf{1}[\mathbf{x}_m \neq \texttt{Mask} \,\&\&\, \mathbf{x}_t = \texttt{Mask}] \odot \log p_\theta(\mathbf{q}, \mathbf{x}_t)[\mathbf{x}_m], \tag{5}$$

which follows from the fact that the model predicts the token distribution for any masked position independently.

# 4 METHODOLOGY

## 4.1 THE FORESEEING DECODING METHOD

Although Equation 3 establishes a simple formulation in decoding $\mathbf{x}_t$, ignoring the contribution of $\mathbf{x}_t$ in $p_{data}(\mathbf{x}_{t+1:T}|\mathbf{q}, \mathbf{x}_{0:t})$ might mislead the decoding process, leading the unexpected errors in the generative response. Therefore, differing from previous works, our proposed Foreseeing Decoding Method (FDM) aims to decode tokens that achieve the largest value across the overall formulation and its decoding strategy can be given as:

$$\pi_F(\mathbf{x}_t|\mathbf{q}, \mathbf{x}_{t-1}) : \mathbf{x}_t = \arg\max_{\mathbf{x}_t} p_\theta(\mathbf{x}_T|\mathbf{q}, \mathbf{x}_t) p_\theta(\mathbf{x}_t|\mathbf{q}, \mathbf{x}_{t-1}). \tag{6}$$

Here we replace $p_{data}(\mathbf{x}_{t+1:T}|\mathbf{q}, \mathbf{x}_{0:t})$ with $p_\theta(\mathbf{x}_T|\mathbf{q}, \mathbf{x}_t)$ owing to the property of the Markov Chain and the modeling of LLDMs. From the theoretical perspective, we also prove that under $\pi_F$, the generative distribution achieves lower KL divergence with the natural distribution $p_{data}$ than that of $\pi_H$:

**Theorem 1.** *Let $\Delta_{total} \triangleq \sum_{t=1}^{T} \mathbb{E}_{p_{data}(\mathbf{x}_{t-1})}[\mathcal{I}_{p_{data}}(\mathbf{x}_t; \mathbf{x}_T|\mathbf{x}_{t-1})]$, where $\mathcal{I}_{p_{data}}(\mathbf{x}_t; \mathbf{x}_T|\mathbf{x}_{t-1})$ is the conditional mutual information under q. Then*

$$D_{KL}(p_{data}(\mathbf{x}), p_{\pi_F}) = D_{KL}(p_{data}(\mathbf{x}), p_{\pi_H}) - \Delta_{total}. \tag{7}$$

For the complete proof, please refer to Appendix B for more details. Due to the fact that mutual information is non-negative, we derive that the generative distribution under $\pi_F$ has lower KL divergence with the natural distribution $p_{data}$ than $\pi_H$. By further applying the log transformation to Equation 6, we can obtain,

$$\mathbf{x}_t = \arg\max_{\mathbf{x}_t} \{\log p_\theta(\mathbf{x}_T|\mathbf{q}, \mathbf{x}_t) + \log p_\theta(\mathbf{x}_t|\mathbf{q}, \mathbf{x}_{t-1})\}. \tag{8}$$

With deeper analysis, we find the first term captures how decoding a specific token influences the future (the **global** confidence, $C_{global}$), while the second term reflects the model's confidence in decoding it at step $t$ (the **local** confidence, $C_{local}$). According to Equation 5, $C_{global}$ can be estimated by:

$$C_{global} = \log p_\theta(\mathbf{x}_T|\mathbf{x}_t, \mathbf{q}) = \mathbf{1}[\mathbf{x}_T \neq \texttt{Mask} \,\&\&\, \mathbf{x}_t = \texttt{Mask}] \odot \log p_\theta(\mathbf{q}, \mathbf{x}_t)[\mathbf{x}_T]. \tag{9}$$

Regarding $\mathbf{x}_T$ is a full response without any mask, $\mathbf{x}_T \neq \texttt{Mask}$ can be omitted since it holds for every position. In addition, instead of greedily decoding $\mathbf{x}_T$ from the model output to calculate $C_{global}$. We compute the expectation over the entire model output distribution:

$$C_{global} = \mathbf{1}[\mathbf{x}_t = \texttt{Mask}] \odot \mathbb{E}_{p_\theta} \log p_\theta(\mathbf{q}, \mathbf{x}_t). \tag{10}$$

---

**Algorithm 1** Foreseeing Decoding Method (FDM)

---

**Input:** User query $\mathbf{q}$, the partially decoded sequence $\mathbf{x}_{t-1}$, pruning threshold $\gamma$, width $K$, well-trained models $\theta$, the number of tokens for decoding $n$.
// Get the candidate tokens of the undecoded positions.
$\mathtt{Candidate} = [\mathbf{1}[\mathbf{x}_{t-1} = \mathtt{Mask}] \odot \arg\max p_\theta(\mathbf{q}, \mathbf{x}_{t-1})]$
**for** $x_t$ in $\mathtt{Candidate}$ **do**
  **if** $p_\theta(\mathbf{q}, \mathbf{x}_{t-1})[x_t] \leq \gamma$ **then**
    Delete $x_t$ from $\mathtt{Candidate}$.
  **end if**
**end for**
Priority Queue $\{\mathbf{x}_t\}$ of the priority $C_{local}$ by foreseeing decoding $n$ tokens from $\mathtt{Candidate}$.
// Narrow the search space with the width $K$.
$\Lambda = \text{Top-K}(\{\mathbf{x}_t\})$
**if** $\Lambda = \varnothing$ **then**
  $\mathbf{x}_t = \arg\max C_{local}(\mathbf{x}_t)$
**else**
  Calculate $C_{global}(\mathbf{x}_t)$ for each $\mathbf{x}_t$ in $\Lambda$
  $\mathbf{x}_t = \arg\max_{\mathbf{x}_t \in \Lambda}\{C_{local}(\mathbf{x}_t) + C_{global}(\mathbf{x}_t)\}$
**end if**
**Output:** The decoded answer $\mathbf{x}_t$ at step $t$

---

In addition, with Equation 5, we can formulate $C_{local}$ with:

$$C_{local} = \mathbf{1}[\mathbf{x}_t \neq \mathtt{Mask} \,\&\&\, \mathbf{x}_{t-1} = \mathtt{Mask}] \odot \log p_\theta(\mathbf{q}, \mathbf{x}_{t-1})[\mathbf{x}_t]. \tag{11}$$

Similar to the denotation in Equation 5, here $[\mathbf{x}_t]$ means the predicted probability of each token in $\mathbf{x}_t$. After combining both equations, we have:

$$\begin{aligned}
\mathbf{x}_t = \arg\max_{\mathbf{x}_t}\{&\mathbf{1}[\mathbf{x}_t = \mathtt{Mask}] \odot \mathbb{E}_{p_\theta} \log p_\theta(\mathbf{q}, \mathbf{x}_t) \\
&+ \mathbf{1}[\mathbf{x}_t \neq \mathtt{Mask} \,\&\&\, \mathbf{x}_{t-1} = \mathtt{Mask}] \odot \log p_\theta(\mathbf{q}, \mathbf{x}_{t-1})[\mathbf{x}_t]\}.
\end{aligned} \tag{12}$$

Note that in this equation, $p_\theta(\mathbf{q}, \mathbf{x}_{t-1})$ is independent of $\mathbf{x}_t$, which can be accurately calculated by querying the model $\theta$ with the acquired sequence $\mathbf{x}_{t-1}$ and user prompt $\mathbf{q}$. In contrast, the $p_\theta(\mathbf{q}, \mathbf{x}_t)$ takes the discrete variable $\mathbf{x}_t$ as a part of the input. It means every evaluation involves a single forward pass. To resolve this problem, we introduce the hyperparameter $K$ to compress the search space for efficiency. In detail, we first obtain the set of candidate tokens with the prediction of $\theta$ on the masked position:

$$\mathtt{Candidate} = \{\mathbf{1}[\mathbf{x}_{t-1} = \mathtt{Mask}] \odot \arg\max p_\theta(\mathbf{q}, \mathbf{x}_{t-1})\} \tag{13}$$

Furthermore, to avoid being trapped in the local optima, we also incorporate a dynamic pruning strategy that retains only candidate tokens whose confidence exceeds the predefined threshold $\gamma$. Then, $C_{local}$ is introduced as the metric to rank each possibility of $\mathbf{x}_t$ and we only keep the Top-$K$ candidates for further decision. Thus, the search space $\Lambda$ can be defined as:

$$\Lambda = \{\mathbf{x}_t | \mathbf{x}_t \in \text{Top-K}(\{\mathbf{x}_t\}), \mathbf{1}[\mathbf{x}_t \neq \mathtt{Mask} \,\&\&\, \mathbf{x}_{t-1} = \mathtt{Mask}] \odot p_\theta(\mathbf{q}, \mathbf{x}_{t-1})[\mathbf{x}_t] > \gamma\}. \tag{14}$$

Defined on $\Lambda$, our proposed **F**oreseeing **D**ecoding **M**ethod (FDM) is formulated as follows,

$$\mathbf{x}_t = \begin{cases} \arg\max_{\mathbf{x}_t \in \{\mathbf{x}_t\}} C_{local}(\mathbf{x}_t) & \text{if } \Lambda = \varnothing \\ \arg\max_{\mathbf{x}_t \in \Lambda}\{C_{local}(\mathbf{x}_t) + C_{global}(\mathbf{x}_t)\} & \text{if } \Lambda \neq \varnothing \end{cases} \tag{15}$$

We also summarize the whole decoding process of FDM at the $t$ step in Algorithm 1.

### 4.2 ACCELERATION WITH THE FORESEEING DECODING METHOD

Based on the analysis in Section 4.1, in this section, we further investigate a variant, *i.e.*, FDM-A to better balance the decoding speed and performance. In Figure 2, we calculate the consistency

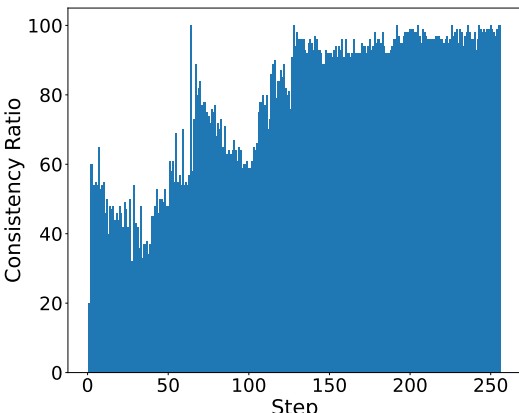

Figure 2: The consistency ratio of selecting the next decoding token using $C_{local}$ alone versus both $C_{local}$ and $C_{global}$. The decisions of both strategies are made based on the same $\mathbf{x}_{t-1}$ in each step. Peak points are observed on the steps of 64 and 128 because we follow the proposed semi-autoregressive pipeline in (Nie et al., 2025) with the block size 64.

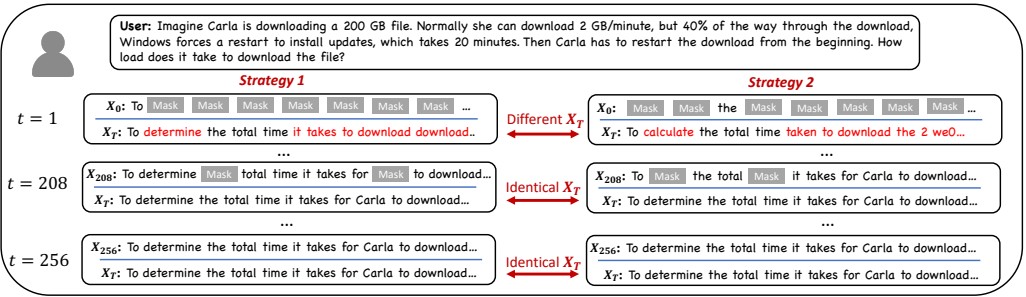

Figure 3: The effect of different decoding strategy to $\mathbf{x}_T$ at the step $t$ given the identical $\mathbf{x}_{t-1}$. The influence gradually decreases as $t$ increases from 0 to T.

ratio in token selection with local confidence only (heuristic decoding) versus both local and global confidence. We average the results over the first 100 examples in the GSM8K (Cobbe et al., 2021) test set. The experiments are performed on LLaDA (Nie et al., 2025), with $\gamma = 0.6$ and $K = 2$.

In the early stages of decoding, due to limited context, the overlap between the two decoding strategies is relatively low, at approximately 50%. However, as decoding progresses, the degree of overlap gradually increases and finally exceeds 90%. It consists of the empirical observation in Figure 3, the final decoding sequence ($\mathbf{X}_T$) is largely shaped when $t$ is small. But when $t$ is approaching $T$, most unmasked tokens have been determined by the adequate contexts, indicating the marginal impact of decoding orders. This evidence demonstrates that more exploration from the global confidence is needed at the beginning, while at the final stage, we only need to adopt the local confidence to accelerate the decoding.

Motivated by this observation, we divide the entire process into three stages: **the exploration, balance and acceleration phases** by incorporating two hyperparameters, *i.e.* $\eta_1$ and $\eta_2$. Firstly, in the exploration phase, if there are no tokens whose prediction probability exceeds $\eta_1$, we will perform exploration with FDM. We denote it with $\mathrm{FDM}_1\left(\mathbf{x}_{t-1}, \mathbf{q}, n\right)$, initialized with the $K_1$. $n$ is the number of tokens decoded in this step and we set it to 1 for reliability. Thus $\mathbf{x}_t$ can be given as:

$$\mathrm{FDM}_1\left(\mathbf{x}_{t-1}, \mathbf{q}, n = 1, \gamma = \gamma_1\right). \tag{16}$$

In the balance phase, we focus on the intermediate states of the decoding process. We combine the exploration and acceleration to achieve a good trade-off. Specifically, we refer to the tokens whose probability lies between $\eta_1$ and $\eta_2$ as the "Borderline Tokens" and those that have the prediction probability greater than $\eta_1$ as the "Qualified Tokens". By applying FDM, we weigh options within a collection comprising both sets. The number of decoding tokens, $n$ is set as $\mathrm{NUM}\left([\mathbf{x}_{t-1} = \mathrm{Mask}]p_\theta(\mathbf{q}, \mathbf{x}_{t-1}) > \eta_1\right)$ for acceleration and $\mathrm{NUM}(\cdot)$ is a function that counts the

---

**Algorithm 2** Acceleration with FDM (FDM-A)

---

**Input:** User query $\mathbf{q}$, fully masked sequence $\mathbf{x}_0$, pruning threshold $\gamma_1$, searching width $K_1$, stage division coefficients $\eta_1$ and $\eta_2$, upper bound for the decoding number $N$, well-trained models $\theta$.
Initialize FDM with $K = K_1$ and get FDM$_1$.
Initialize FDM with $K = 1$ and get FDM$_2$.
$t = 1$
**while** Mask not in $\mathbf{x}_t$ **do**
   **if** NUM$\big([\mathbf{x}_{t-1} = \text{Mask}]p_\theta(\mathbf{q}, \mathbf{x}_{t-1}) > \eta_1\big) = 0$ **then**
      $\mathbf{x}_t = \text{FDM}_1(\mathbf{x}_{t-1}, \mathbf{q}, n = 1, \gamma = \gamma_1)$
   **else if** NUM$\big([\mathbf{x}_{t-1} = \text{Mask}]p_\theta(\mathbf{q}, \mathbf{x}_{t-1}) > \eta_1\big) \geq N$ **then**
      $\mathbf{x}_t = \text{FDM}_2(\mathbf{x}_{t-1}, \mathbf{q}, n = N, \gamma = 1.0)$
   **else if** NUM$\big(\eta_2 < [\mathbf{x}_{t-1} = \text{Mask}]p_\theta(\mathbf{q}, \mathbf{x}_{t-1}) \leq \eta_1\big) = 0$ **then**
      $\mathbf{x}_t = \text{FDM}_2(\mathbf{x}_{t-1}, \mathbf{q}, n = \text{NUM}([\mathbf{x}_{t-1} = \text{Mask}]p_\theta(\mathbf{q}, \mathbf{x}_{t-1}) > \eta_1), \gamma = 1.0)$
   **else**
      $\mathbf{x}_t = \text{FDM}_1(\mathbf{x}_{t-1}, \mathbf{q}, n = \text{NUM}([\mathbf{x}_{t-1} = \text{Mask}]p_\theta(\mathbf{q}, \mathbf{x}_{t-1}) > \eta_1), \gamma = \eta_2)$
   **end if**
   $t = t + 1$
**end while**
**Output:** Generated answer $\mathbf{x}_t$.

---

number of tokens that meet the given condition. The formulation of decoding $\mathbf{x}_t$ is:

$$\text{FDM}_1\Big(\mathbf{x}_{t-1}, \mathbf{q}, n = \text{NUM}\big([\mathbf{x}_{t-1} = \text{Mask}]p_\theta(\mathbf{q}, \mathbf{x}_{t-1}) > \eta_1\big), \gamma = \eta_2\Big), \tag{17}$$

As shown in Figure 3, different decoding methods perform almost the same at the tail, indicating the convergence in decoding. At this time, we shift the mode to the acceleration phase. The tokens are decoded with the local confidence only to achieve the fastest speed. The decoding function can be regarded as a special case of FDM. By initializing it with $K = 1$, we represent it by $\text{FDM}_2(\mathbf{x}_{t-1}, \mathbf{q}, n)$ and $\mathbf{x}_t$ can be decoded with:

$$\text{FDM}_2\Big(\mathbf{x}_{t-1}, \mathbf{q}, n = \min\big(\text{NUM}([\mathbf{x}_{t-1} = \text{Mask}]p_\theta(\mathbf{q}, \mathbf{x}_{t-1}) > \eta_1), N\big), \gamma = 1.0\Big). \tag{18}$$

$N$ is the upper bound for clipping the decoding number of tokens.

We term this acceleration decoding pipeline as FDM-A. The algorithm of it is summarized in Algorithm 2.

## 5 EXPERIMENT

### 5.1 MAIN RESULTS

**Benchmark and Metrics:** To demonstrate the effectiveness of FDM and FDM-A, we conduct experiments on four prevailing benchmark datasets: GSM8K (Cobbe et al., 2021), HumanEval (Chen et al., 2021), Countdown (Zhao et al., 2025) and ARC (Clark et al., 2018). They reflect the capabilities of LLDMs in four aspects, including math issue solutions, code generation, logic reasoning, and common sense knowledge. Meanwhile, for the convenience of evaluation, following (Hong et al., 2025), we add a system prompt before each input item. The details of them in each dataset are summarized in Appendix C. All evaluations are performed under the zero-shot condition to ensure the fairness. For performances, we take accuracy as the metric, manifesting the percentage of queries, that could be properly answered by LLDMs. For efficiency, we choose Tokens Per Second (TPS) as the metric. It is defined as the number of tokens generated in one second.

**Baselines and models:** We compare FDM with three heuristic-based decoding methods: decoding with the highest probability (Probability) (Nie et al., 2025; Zheng et al., 2024), with the highest marginal probability (Margin) (Kim et al., 2025) and with the lowest entropy (Entropy) (Ben-Hamu et al., 2025). For FDM-A, since it tries to balance between efficiency and performance, we also compare it with the newest dynamic decoding methods for LLDMs in recent months: EB (Entropy Bounded Sampler) (Ben-Hamu et al., 2025) and WINO (Hong et al., 2025). To ensure

Table 2: Comparison (%) of FDM with the heuristic decoding strategies across four benchmarks. With the scale up of the wide $K$, we see that accuracy increases with the decrease of TPS, demonstrating that FDM serves as an inference-time scaling method.

| Benchmark | Method | LLaDA | | LLaDA-1.5 | | MMaDA-MixCoT | | LLaDA-MoE | |
|---|---|---|---|---|---|---|---|---|---|
| | | Accuracy | TPS | Accuracy | TPS | Accuracy | TPS | Accuracy | TPS |
| GSM8K | Probability ($T$=256) | 81.20 | 11.51 | 82.10 | 11.89 | 56.18 | 11.26 | 76.50 | 4.13 |
| | Margin ($T$=256) | 80.14 | 11.23 | 82.18 | 11.39 | 56.18 | 10.97 | 76.80 | 4.04 |
| | Entropy ($T$=256) | 80.12 | 10.79 | 81.93 | 10.94 | 54.91 | 10.51 | 76.80 | 3.84 |
| | FDM (K=2) | 82.03 | 8.28 | 82.49 | 8.29 | 57.54 | 8.18 | 77.48 | 3.80 |
| | FDM (K=3) | 82.18 | 5.86 | 82.64 | 5.84 | 57.68 | 5.71 | 77.63 | 3.78 |
| | FDM (K=4) | 82.34 | 4.61 | 83.02 | 4.69 | 57.92 | 4.61 | 78.32 | 3.60 |
| HumanEval | Probability ($T$=256) | 42.68 | 9.24 | 42.68 | 9.20 | 12.80 | 8.91 | 56.71 | 3.81 |
| | Margin ($T$=256) | 43.29 | 8.76 | 42.68 | 8.83 | 12.80 | 8.65 | 58.54 | 3.81 |
| | Entropy ($T$=256) | 40.24 | 8.67 | 42.68 | 8.67 | 11.59 | 8.28 | 59.15 | 3.56 |
| | FDM (K=2) | 43.29 | 6.55 | 43.29 | 6.66 | 12.80 | 6.32 | 59.76 | 3.72 |
| | FDM (K=3) | 44.51 | 4.63 | 43.29 | 4.51 | 12.80 | 4.60 | 59.76 | 3.67 |
| | FDM (K=4) | 45.73 | 3.66 | 44.51 | 3.71 | 13.40 | 3.64 | 60.37 | 3.47 |
| Countdown | Probability ($T$=256) | 18.75 | 11.19 | 16.02 | 11.20 | 4.69 | 11.15 | 40.62 | 4.15 |
| | Margin ($T$=256) | 19.14 | 10.89 | 20.31 | 10.92 | 14.61 | 10.84 | 40.23 | 4.03 |
| | Entropy ($T$=256) | 18.44 | 10.39 | 20.31 | 10.39 | 7.03 | 10.31 | 38.67 | 3.95 |
| | FDM (K=2) | 19.14 | 8.21 | 20.70 | 8.62 | 14.45 | 8.13 | 40.62 | 3.73 |
| | FDM (K=3) | 21.09 | 6.11 | 21.88 | 6.48 | 16.40 | 6.10 | 46.10 | 3.71 |
| | FDM (K=4) | 25.00 | 5.28 | 23.43 | 5.25 | 17.58 | 4.97 | 46.88 | 3.70 |
| ARC | Probability ($T$=256) | 80.96 | 10.96 | 87.06 | 10.98 | 58.26 | 10.94 | 76.19 | 4.01 |
| | Margin ($T$=256) | 82.55 | 10.85 | 87.44 | 10.82 | 56.37 | 10.62 | 75.85 | 4.01 |
| | Entropy ($T$=256) | 78.13 | 10.43 | 87.38 | 10.38 | 58.33 | 10.18 | 71.84 | 3.98 |
| | FDM (K=2) | 86.00 | 7.72 | 88.18 | 7.71 | 59.43 | 7.68 | 82.89 | 3.85 |
| | FDM (K=3) | 86.46 | 5.58 | 88.45 | 5.59 | 59.61 | 5.58 | 83.45 | 3.73 |
| | FDM (K=4) | 86.68 | 4.58 | 88.59 | 4.56 | 59.76 | 4.37 | 83.51 | 3.64 |

the generalability of the observations, we cover four variants of LLDMs: LLaDA-8B-Instruct (Nie et al., 2025), LLaDA-1.5 (Zhu et al., 2025), LLaDA-MoE-7B-Instruct (inclusionAI, 2025) and MMaDA-8B-MixCoT (Yang et al., 2025).

**Hyperparameters:** Consistent with the observation in (Nie et al., 2025), we observe that the semi-autoregressive pipeline is vital to maintain a satisfying performance for the instruct models of LLDMs. Thus we applying it for decoding with the generation length 256, block size 64. The step number $T$ is fixed to 256 and 128, respectively for the heuristic decoding methods when comparing them with FDM or FDM-A. The threshold in EB is set as 0.5. $\tau_1$ and $\tau_2$ in WINO are configured as 0.7 and 0.9. For our method, $\gamma$ for the dynamic pruning is 0.6 for FDM and 0.5 for FDM-A considering explorations are less performed in FDM-A. For the threshold for stage division, we set $\eta_1$ to 0.8 and $\eta_2$ to 0.7. We gradually increase the width of FDM from 2 to 4 and set the default width of FDM-A as 2. All experiments are performed on a single NVIDIA A100 80G GPU.

**Results:** Firstly, in Table 2, we observe that FDM surpasses baseline methods across all models and benchmarks, demonstrating the significance of exploration with the global confidence. For example, on the LLaDA model and ARC benchmarks, the highest score of the heuristic methods is 82.55%, and FDM with the search width 2 is 86.00%. In addition, scaling up the width $K$ can further enhance its performance, demonstrating its role in serving as an inference-time scaling technique. An example is that on the LLaDA-MoE and GSM8K benchmark, the accuracy improves from 77.48% to 78.32% when the width rises from 2 to 4. It is worth noting that this improvement is notable since we do not need a verifier or optimize the parameters. It demonstrates that FDM well fits the scenarios that have sufficient computations but need high accuracy. In Appendix D, we show an example case that could be properly decoded with FDM but incorrectly decoded for all heuristic methods.

Secondly, in Table 3, the results reveal that FDM-A achieves an outstanding trade-off between accuracy and speed: When we compare FDM-A's performance with FDM, the negative influence is marginal. The accuracy of FDM ($K = 2$) on the GSM8K benchmark of the LLaDA model is 82.03%, and for FDM-A, it achieves 81.96% (-0.07%). Meanwhile, FDM-A achieves 5.15× speedup in TPS, significantly improving TPS from 8.28 to 42.65. In contrast, the performance of heuristic methods will largely degrade when their decoding step is reduced. For example, when we compare the results in Table 2 and 3, the accuracy with the highest probability decoding declines from 81.20% to 78.17%

Table 3: Comparison (%) of FDM-A with methods that accelerate the decoding of LLDMs. The best results are in **bold**. FDM-A achieves not only the highest accuracy, but also the fastest speed.

| Benchmark | Method | LLaDA | | LLaDA-1.5 | | MMaDA-MixCoT | | LLaDA-MoE | |
|---|---|---|---|---|---|---|---|---|---|
| | | Accuracy | TPS | Accuracy | TPS | Accuracy | TPS | Accuracy | TPS |
| GSM8K | Probability ($T$=128) | 78.17 | 20.86 | 80.06 | 22.04 | 52.08 | 21.65 | 71.49 | 8.81 |
| | Margin ($T$=128) | 79.23 | 21.36 | 80.39 | 21.61 | 50.72 | 21.37 | 72.93 | 8.47 |
| | Entropy ($T$=128) | 77.94 | 20.69 | 80.22 | 20.71 | 49.13 | 20.51 | 70.66 | 8.39 |
| | EB | 79.61 | 36.46 | 81.35 | 37.74 | 53.83 | 36.39 | 74.53 | 13.98 |
| | WINO | 79.45 | 40.02 | 80.89 | 40.96 | 51.33 | 36.92 | 72.78 | 13.02 |
| | FDM-A (Ours) | **81.96** | **42.65** | **82.87** | **43.98** | **55.12** | **40.96** | **76.72** | **14.89** |
| HumanEval | Probability ($T$=128) | 35.37 | 16.75 | 39.63 | 17.04 | 7.93 | 16.59 | 51.22 | 7.99 |
| | Margin ($T$=128) | 37.20 | 16.25 | 37.20 | 16.58 | 12.80 | 16.15 | 49.39 | 7.74 |
| | Entropy ($T$=128) | 31.20 | 15.93 | 35.37 | 15.77 | 4.27 | 15.46 | 48.78 | 7.46 |
| | EB | 37.20 | 17.34 | 37.80 | 18.99 | 12.80 | 24.95 | 59.15 | 10.14 |
| | WINO | 39.02 | 18.33 | 40.24 | 20.59 | 12.20 | 29.90 | 56.71 | 9.31 |
| | FDM-A (Ours) | **44.51** | **21.56** | **42.07** | **23.83** | **13.41** | **32.32** | **60.98** | **11.25** |
| Countdown | Probability ($T$=128) | 19.53 | 21.19 | **21.09** | 22.01 | 12.01 | 21.97 | 44.53 | 8.80 |
| | Margin ($T$=128) | 20.31 | 20.65 | 19.92 | 21.76 | 8.59 | 21.57 | 42.58 | 8.33 |
| | Entropy ($T$=128) | 19.53 | 20.62 | 18.75 | 20.93 | 8.59 | 20.52 | 35.94 | 8.15 |
| | EB | 19.20 | 18.68 | 19.92 | 18.79 | 18.75 | 48.51 | 42.19 | 9.48 |
| | WINO | 19.14 | 20.52 | 19.14 | 20.38 | 20.70 | 52.39 | 25.00 | 8.23 |
| | FDM-A (Ours) | **21.48** | **21.98** | **21.09** | **22.29** | **26.95** | **62.30** | **49.61** | **10.14** |
| ARC | Probability ($T$=128) | 82.16 | 21.24 | 86.38 | 21.71 | 55.04 | 22.26 | 72.74 | 8.68 |
| | Margin ($T$=128) | 84.34 | 20.65 | 85.91 | 21.34 | 56.30 | 21.47 | 75.48 | 8.56 |
| | Entropy ($T$=128) | 77.20 | 19.91 | 84.58 | 20.48 | 55.61 | 20.58 | 69.08 | 8.12 |
| | EB | 73.55 | 32.01 | 85.68 | 34.89 | 56.86 | 32.89 | 71.50 | 8.98 |
| | WINO | 79.44 | 34.17 | 85.31 | 35.27 | 56.79 | 34.27 | 70.86 | 7.92 |
| | FDM-A (Ours) | **86.30** | **38.20** | **87.66** | **38.43** | **59.50** | **37.07** | **83.33** | **12.62** |

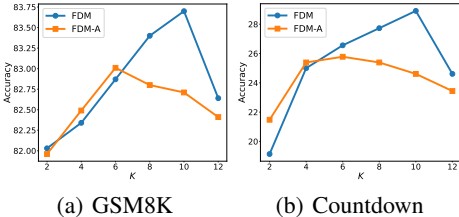

Figure 4: The influence of $K$ to model performance on GSM8K and Countdown benchmarks.

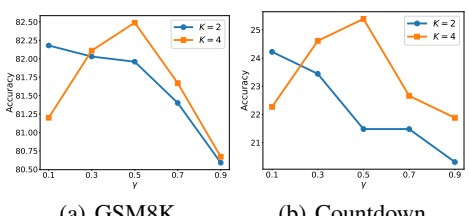

Figure 5: The influence of $\gamma$ to model performance on GSM8K and Countdown benchmarks.

when we halve the step number. Owing to its full exploitation with the global confidence in the exploration and balance stage, FDM-A also outperforms dynamic decoding methods like WINO, demonstrating its strong capacity.

## 5.2 ABLATION STUDIES

**Going Wider with $K$.** In Section 5.1, we show that larger $K$ will benefit the performance of FDM and a natural question is whether it will bring consistent improvement with the increase of $K$. Thus, here we set $K$ with the value from $[2, 4, 6, 8, 10, 12]$ and perform experiments on the LLaDA model. As summarized in both Figure 4 and Figure 8 in Appendix E, the accuracy will reach the peak for both FDM-A and FDM. This is because although the training goal of the network is to fit the data distribution $p_{data}$, in realistic, the divergence between two distributions is not zero, leading to accumulating bias when we perform excessive searching operations, supported by the theoretical analysis in Appendix F. Intriguingly, when we compare the curves of FDM and FDM-A, we find that FDM outperforms FDM-A when $K$ is larger. Conversely, FDM-A holds an advantage when $K$ is small. This highlights the complementary role of both methods. FDM-A is suited for use when computational resources are limited. FDM, on the other hand, provides robust capabilities under conditions of abundant computational supply.

**The Setting of $\gamma$.** When we foresee the future impact with the global confidence, we introduce a threshold $\gamma$ to dynamically prune the candidate tokens of the low local confidence value. We further investigate its effect by tuning it between 0.1 and 0.9 with $K = 2$ and 4. Taking FDM-A as an example, the results on LLaDA of the GSM8K and Countdown datasets are shown in Figure 5. For

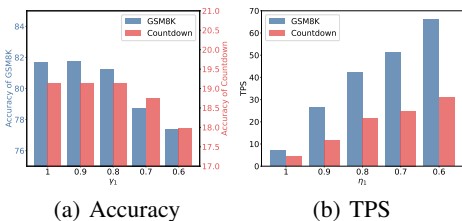 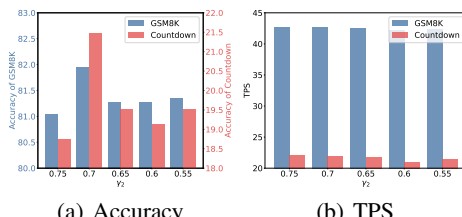

|  |  |  |  |
|---|---|---|---|
| (a) Accuracy | (b) TPS | (a) Accuracy | (b) TPS |

Figure 6: The influence of $\eta_1$ to Accuracy and TPS on GSM8K and Countdown benchmarks.

Figure 7: The influence of $\eta_2$ to Accuracy and TPS on GSM8K and Countdown benchmarks.

more results, please refer to Figure 9 in Appendix E. We also observe a trade-off in its configuration: the results illustrate that a smaller $\gamma$, *e.g.* 0.1, achieves better accuracy in $K = 2$. This is because it provides a wider choice for selection, mitigating the issue of inadequate search. However, we find that it can not maintain its advantage when $K$ increases to 4, since more tokens of lower $C_{local}$ are chosen, reflecting the uncertainty of models in its prediction. In contrast, if we set $\gamma$ too large, it will suppress the exploration in the foreseeing decoding. In total, $\gamma$ near 0.5 can balance both in its application, which is chosen as the configuration for the experiments in Section 5.1.

**The Choices of $\eta_1$ and $\eta_2$.** In addition to $K$ and $\gamma$, FDM-A also introduces $\eta_1$ and $\eta_2$ as hyperparameters for stage divisions. In the main text, we perform experiments on the GSM8K and Countdown benchmarks in Figure 6 and 7. For more results on HumanEval and ARC benchmarks, please refer to Figure 10 and 11 in Appendix E. We firstly fix $\eta_2$ at 0.6 and linearly decrease $\eta_1$ from 1.0 to 0.6. The results in Figure 6 (a) demonstrate that the accuracy remains stable at first, and then drops sharply in small $\eta_1$. In contrast, the TPS monotonically improves with the decrease of $\gamma_1$ because more tokens are parally decoded. By default, we set $\gamma_1$ as 0.8 because it can not only maintain the model utility but also achieve high decoding speed. Secondly, we fix $\eta_1$ with 0.8 and set $\eta_2$ from [0.75, 0.7, 0.65, 0.6, 0.55]. Among all configurations, $\eta_2 = 0.7$ consistently achieves outstanding performances across all benchmarks. Because when $\eta_2$ is large, the exploration at the balance stage is insufficient. But if $\eta_2$ is set too small, uncertain tokens will interfere the correctness of decoding.

## 6 CONCLUSION

In this paper, we address the critical challenge of token decoding orders in Large Language Diffusion Models (LLDMs). We identify the limitations of existing heuristic methods that rely solely on local confidence, and propose a novel solution by incorporating global confidence into the decoding process. We theoretically prove that it will achieve a lower value than heuristic methods in divergence with the natural distribution. To save the computations, our proposed Foreseeing Decoding Method (FDM) optimizes decoding order through a heuristic beam search, effectively balancing local and global considerations. The accelerated variant, FDM-A, further enhances efficiency by strategically applying deep exploration only at critical steps. Extensive experiments demonstrate that FDM consistently outperforms existing baselines, serving as an effective inference-time scaling method, while FDM-A achieves an outstanding trade-off between performance and speed. Our work provides a principled approach to decoding strategy design, potentially opening new avenues for developing more powerful and efficient decoding methods for LLDMs.

## ETHICS STATEMENT

Since this work is dedicated to enhancing the efficiency and accuracy of LLDM by proposing decoding algorithms, its technical focus inherently does not involve the creation or propagation any contents of ethical issues. In addition, our research utilizes exclusively publicly available datasets and models for experiments and analysis, with all sources properly cited.

## REPRODUCIBILITY STATEMENT

The experimental settings are elaborated in detail in Section 5 and Appendix C. We will also release the source code, along with the necessary configuration files, upon acceptance.

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

## A    USAGE OF LLM

In this paper, we employ Large Language Models (LLMs) to polish writing and assist in debugging. All LLM-polished text is rechecked by the authors to ensure accuracy and prevent over-claims or confusions.

## B    PROOF OF THEOREM 1

Define the single-step KL errors

$$\varepsilon_t^H \triangleq D_{KL}(p_{data}(\mathbf{x}_t|\mathbf{x}_{t-1}), \pi_H(\mathbf{x}_t)), \qquad \varepsilon_t^F \triangleq D_{KL}(p_{data}(\mathbf{x}_t|\mathbf{x}_{t-1}), \pi_F(\mathbf{x}_t)).$$

We first prove that

$$\varepsilon_t^F = \varepsilon_t^H - \mathcal{I}_{p_{data}}(\mathbf{x}_t; \mathbf{x}_T|\mathbf{x}_{t-1}), \tag{19}$$

where $\mathcal{I}_{p_{data}}(\mathbf{x}_t; \mathbf{x}_T|\mathbf{x}_{t-1})$ is the conditional mutual information under $q$. Since $\mathbf{x}_{t-1}$ and $\mathbf{q}$ can be considered as fixed variables at the step $t$, we omit them in the notation for brevity. According to the definition of KL divergence, we have:

$$\varepsilon_t^F \triangleq D_{KL}\big(p_{data}(\mathbf{x}_t), \pi_F(\mathbf{x}_t)\big) = \sum_{\mathbf{x}_t} p_{data}(\mathbf{x}_t)\Big[\log p_{data}(\mathbf{x}_t) - \log \pi_F(\mathbf{x}_t)\Big]. \tag{20}$$

By construction

$$\pi_F(\mathbf{x}_t) = \frac{\exp(S(\mathbf{x}_t))}{Z_t}, \qquad S(\mathbf{x}_t) = C_{\text{local}}(\mathbf{x}_t) + C_{\text{global}}(\mathbf{x}_t), \qquad Z_t = \sum_{\mathbf{x}_t'} \exp(S(\mathbf{x}_t')).$$

Hence

$$\log \pi_F(\mathbf{x}_t) = S(\mathbf{x}_t) - \log Z_t = C_{\text{local}}(\mathbf{x}_t) + C_{\text{global}}(\mathbf{x}_t) - \log Z_t. \tag{21}$$

By inserting Equation 21 into 20 and split the sum,

$$\varepsilon_t^F = \sum_{\mathbf{x}_t} p_{data}(\mathbf{x}_t)\Big[\log p_{data}(\mathbf{x}_t) - C_{\text{local}}(\mathbf{x}_t) - C_{\text{global}}(\mathbf{x}_t) + \log Z_t\Big]$$

$$= \underbrace{\sum_{\mathbf{x}_t} p_{data}(\mathbf{x}_t)\Big[\log p_{data}(\mathbf{x}_t) - C_{\text{local}}(\mathbf{x}_t)\Big]}_{\text{Term A}} - \underbrace{\sum_{\mathbf{x}_t} p_{data}(\mathbf{x}_t)\Big[C_{\text{global}}(\mathbf{x}_t) - \log Z_t\Big]}_{\text{Term B}}. \tag{22}$$

By definition $C_{\text{local}}(\mathbf{x}_t) = \log p_\theta(\mathbf{x}_t)$, so

$$\text{Term A} = \sum_{\mathbf{x}_t} p_{data}(\mathbf{x}_t)\Big[\log p_{data}(\mathbf{x}_t) - \log p_\theta(\mathbf{x}_t)\Big] = D_{KL}\big(p_{data}(\mathbf{x}_t), p_\theta(\mathbf{x}_t)\big) \triangleq \varepsilon_t^H. \tag{23}$$

First recall

$$C_{\text{global}}(\mathbf{x}_t) = \mathbb{E}_{x' \sim p_\theta} \log p_\theta(x' \mid \mathbf{x}_t) = \sum_{\mathbf{x}_T} p_\theta(\mathbf{x}_T \mid \mathbf{x}_t) \log p_\theta(\mathbf{x}_T \mid \mathbf{x}_t).$$

Therefore

$$\text{Term B} = \sum_{\mathbf{x}_t} p_{data}(\mathbf{x}_t)\Big[C_{\text{global}}(\mathbf{x}_t) - \log Z_t\Big]$$

$$= \sum_{\mathbf{x}_t} p_{data}(\mathbf{x}_t)\Big[\sum_{\mathbf{x}_T} p_\theta(\mathbf{x}_T \mid \mathbf{x}_t) \log p_\theta(\mathbf{x}_T \mid \mathbf{x}_t) - \log Z_t\Big]. \tag{24}$$

Next we use the identity $Z_t = p_\theta(\mathbf{x}_T)$ (marginal over $\mathbf{x}_t'$), which follows from

$$Z_t = \sum_{\mathbf{x}_t'} \exp(S(\mathbf{x}_t')) = \sum_{\mathbf{x}_t'} p_\theta(\mathbf{x}_t', \mathbf{x}_T) = p_\theta(\mathbf{x}_T).$$

Hence

$$\log Z_t = \log p_\theta(\mathbf{x}_T) = \sum_{\mathbf{x}_T} p_\theta(\mathbf{x}_T \mid \mathbf{x}_t) \log p_\theta(\mathbf{x}_T),$$

where the second equality holds because $p_\theta(\mathbf{x}_T)$ does not depend on $\mathbf{x}_t$ and $\sum_{\mathbf{x}_t} p_\theta(\mathbf{x}_T \mid \mathbf{x}_t) = 1$. Inserting this into Equation 24 gives

$$\text{Term B} = \sum_{\mathbf{x}_t} p_{data}(\mathbf{x}_t) \sum_{\mathbf{x}_T} p_\theta(\mathbf{x}_T \mid \mathbf{x}_t) \Big[ \log p_\theta(\mathbf{x}_T \mid \mathbf{x}_t) - \log p_\theta(\mathbf{x}_T) \Big]$$

$$= \sum_{\mathbf{x}_t, \mathbf{x}_T} p_{data}(\mathbf{x}_t, \mathbf{x}_T) \Big[ \log \frac{p_\theta(\mathbf{x}_T \mid \mathbf{x}_t)}{p_\theta(\mathbf{x}_T)} \Big]$$

$$= \sum_{\mathbf{x}_t, \mathbf{x}_T} p_{data}(\mathbf{x}_t, \mathbf{x}_T) \Big[ \log \frac{p_{data}(\mathbf{x}_t, \mathbf{x}_T)}{p_{data}(\mathbf{x}_t) p_{data}(\mathbf{x}_T)} \Big] \qquad \text{(replace } p_\theta \text{ with } q \text{ inside log)}$$

$$= \mathcal{I}_{p_{data}}(\mathbf{x}_t; \mathbf{x}_T). \tag{25}$$

Inserting Equation 23 and 25 into Equation 22 yields

$$\varepsilon_t^F = \varepsilon_t^H - \mathcal{I}_{p_{data}}(\mathbf{x}_t; \mathbf{x}_T),$$

which completes the proof of Equation 19. Using the chain rule for KL divergence over sequences we have

$$D_{KL}\big(p_{data}(\mathbf{x}), p_{\pi_F}\big) = \sum_{t=1}^{T} \mathbb{E}_{p_{data}(\mathbf{x}_{t-1})}\big[\varepsilon_t^F\big] = \sum_{t=1}^{T} \mathbb{E}_{p_{data}(\mathbf{x}_{t-1})}\big[\varepsilon_t^H - \mathcal{I}_{p_{data}}(\mathbf{x}_t; \mathbf{x}_T | \mathbf{x}_{t-1})\big].$$

Recognising the definitions

$$\sum_{t=1}^{T} \mathbb{E}_{p_{data}(\mathbf{x}_{t-1})} \varepsilon_t^H = D_{KL}\big(p_{data}(\mathbf{x}), p_{\pi_H}\big), \quad \sum_{t=1}^{T} \mathbb{E}_{p_{data}(\mathbf{x}_{t-1})} \mathcal{I}_{p_{data}}(\mathbf{x}_t; \mathbf{x}_T | \mathbf{x}_{t-1}) = \Delta_{\text{total}},$$

we obtain

$$D_{KL}\big(p_{data}(\mathbf{x}), p_{\pi_F}\big) = D_{KL}\big(p_{data}(\mathbf{x}), p_{\pi_H}\big) - \Delta_{\text{total}},$$

which finishes the proof of Theorem 1. $\qquad\square$

## C   SYSTEM PROMPTS FOR EVALUATION

---

**GSM8K**

You are a math expert. You will be given a question to solve. Solve it step by step. Wrap the final answer in a \\boxed{}.
Respond in the following format:
<reasoning >
Your reasoning here
</reasoning >
<answer>
\\boxed{...}
</answer>

---

**ARC**

You will be given a multiple choice question and a list of options. You need to first reason step by step, and then select the correct option (A, B, C, D). Wrap the single letter of the correct option in \\boxed{...}. Respond in the following format:
<reasoning>
Your reasoning here
</reasoning>
<answer>
\\boxed{...}
</answer>

---

**Countdown**

Using only the provided numbers, create an arithmetic expression that evaluates to exactly the provided target number. You may use the operations +, -, *, and / as needed, but each number must be used exactly once. Think step-by-step. After reasoning, provide only your final expression inside \\boxed{}
tags without including an equals sign or the target number. For example: \\boxed{a + b * c}
Respond in the following format:
<reasoning>
Your reasoning here
</reasoning>
<answer>
\\boxed{...}
</answer>

**HumanEval**

Write a solution to the following problem and make sure that it passes the tests:
python

## D  THE EXAMPLE CASE

**Question in the ARC benchmark**

Which of the following is the primary way that evergreen trees planted on the north and west sides of residences in Massachusetts help reduce winter heating costs?
A  They insulate against cold air
B  They optimize solar heating.
C  They reduce radiant cooling.
D  They act as a windbreak.

**Decoding with the highest probability (Incorrect)**

Evergreen trees planted on the north and west sides of residences in Massachusetts help reduce winter heating costs primarily by acting as a physical barrier against cold cold air. This trees' asymmetrical and dense foliage provide a a natural insulation, preventing cold air from entering the home, thereby reducing the need for heating heating.
</reasoning>
<answer>
\boxed{A}
</answer>

**Decoding with the highest margin (Incorrect)**

Evergreen trees planted on the north and west sides of residences in Massachusetts can help reduce winter heating costs by providing as much solar heat as possible. This is because the trees can capture and store sunlight, which can be used to heat the home during the winter months. By maximizing the amount of solar heat, the trees can help reduce the need for additional heating systems.
</reasoning>

<answer>
B

---

**Decoding with the lowest entropy (Incorrect)**

Evergreen trees are planted on the north and west sides of residences in Massachusetts to help reduce winter heating costs. The primary reason for this benefit is due to the insulation provided by the trees. The trees provide insulation against the cold, reducing the amount of heat escaping the house and keeping the interior temperature slightly warmer. This insulation reduces the need for heating, thereby making the home more efficient.
<answer>
\boxed{A}

---

**Decoding with FDM (Correct)**

Evergreen trees planted on the north and west sides of residences in Massachusetts can help reduce winter heating costs by acting as a windbreak. These trees can block cold air from entering the home, reducing the need for heating. Additionally, evergreen trees can also provide insulation against cold air, but the primary benefit is their role as a windbreak.
</reasoning>
<answer>
\boxed{D}
</answer>

---

# E  ABLATION STUDIES ON THE HUMANEVAL AND ARC BENCHMARKS

We perform experiments on the HumanEval and ARC benchmarks with the experimental settings in Section 5.2. We observe that the optimal hyperparameter has good transferability across datasets.

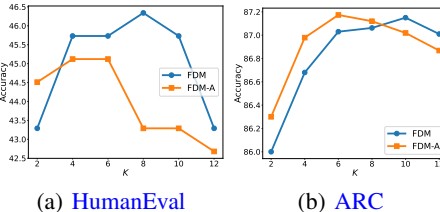

(a) HumanEval  (b) ARC

Figure 8: The influence of $K$ to model performance on HumanEval and ARC benchmarks.

(a) HumanEval  (b) ARC

Figure 9: The influence of $\gamma$ to model performance on HumanEval and ARC benchmarks.

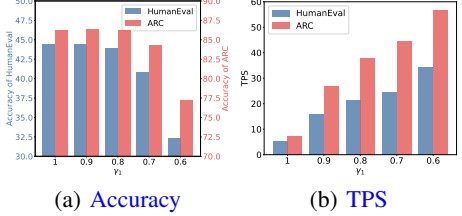

(a) Accuracy  (b) TPS

Figure 10: The influence of $\eta_1$ to Accuracy and TPS on HumanEval and ARC benchmarks.

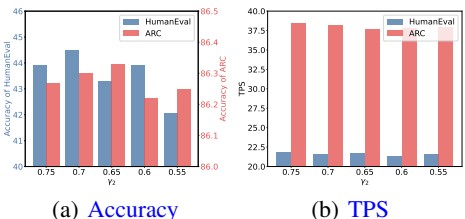

(a) Accuracy  (b) TPS

Figure 11: The influence of $\eta_2$ to Accuracy and TPS on HumanEval and ARC benchmarks.

# F  THEORETICAL ANALYSIS FOR PERFORMANCE DEGRADATION WITH A EXTREMELY LARGE $K$

We next present a theoretical analysis to support that the degradation in performance is due to the consequence of noise-amplified selection in the mismatch between the model and natural distributions. At each decoding step, we have $K$ candidate tokens whose true future returns are

$$s_1, \ldots, s_K \in \mathbb{R}, \quad s_* = \max_i s_i, \text{where } s_i \triangleq \log p_{\text{data}}(\mathbf{x}_t = \text{i-th candidate token} \mid \mathbf{x}_{t-1}, \mathbf{q}).$$

When $K$ is large enough, we can assume that the token with the highest possibility in the real data distribution is already in the candidates. The model in our algorithm only observes confidence scores, which are noisy estimates of true future returns:

$$\hat{s}_i = s_i + \xi_i$$

Here we hypothesize that $\xi_i$ is independent and identically distributed (i.i.d.) Gaussian random variables:

$$\xi_i \overset{\text{iid}}{\sim} \mathcal{N}(0, \sigma^2)$$

and $\hat{s}_i$ is equal to the sum of the local and global confidence of the i-th candidate token:

$$\hat{s}_i = C_{local}(\text{i-th candidate token}) + C_{global}(\text{i-th candidate token})$$

According to Equation 8, our algorithm selects the candidate with the highest $\hat{s}_i$:

$$j = \arg\max_{i \in [K]} \hat{s}_i.$$

For any sub-optimal candidate $i$ with confidence gap $\Delta_i = s_* - s_i > 0$, the probability that noise flips the ranking is

$$\Pr(\hat{s}_i > \hat{s}_*) = \Pr(\xi_i - \xi_* > \Delta_i) = \Phi\left(-\frac{\Delta_i}{\sqrt{2}\sigma}\right),$$

where $\Phi(\cdot)$ is the standard normal tail CDF(Cumulative Distribution Function). Union-bounding over the $K - 1$ sub-optimal candidates yields

$$\Pr(\text{select non-optimal}) \leq \sum_{i \neq *} \Phi\left(-\frac{\Delta_i}{\sqrt{2}\sigma}\right).$$

The RHS(right hand side) increases monotonically with $K$ whenever there exist $\Delta_i = O(\sigma)$. On the other hand, let $\xi_{(K)} = \max_{i \leq K} \xi_i$. The Extreme-value Theory gives

$$\mathbb{E}[\xi_{(K)}] \approx \sigma\sqrt{2\ln K}.$$

Hence, the selected estimate satisfies

$$\mathbb{E}[\hat{s}_j] \approx s_* + \sigma\sqrt{2\ln K},$$

while the true return of the selected candidate satisfies:

$$\mathbb{E}[s_j] \leq s_* + \underbrace{\mathbb{E}[\xi_{(K)} - \xi_j]}_{\geq 0}.$$

Consequently, the expected regret is:

$$\mathbb{E}[\Delta_j] = \mathbb{E}[s_* - s_j] \geq C\sigma\sqrt{\ln K}$$

which grows with $K$, providing a quantitative lower bound for the "winner's curse" in the decoding process. Because each incorrect selection changes the subsequent context, the regret accumulates along the generation process. Early mistakes (where $K$ is too large in our schedule) therefore have a negative effect, explaining the empirical observation in Figure 4 and 8.

