# OpenReview forum: "Decoding Large Language Diffusion Models with Foreseeing Movement"
_ICLR.cc/2026/Conference — Submitted to ICLR 2026_

### Official Review · Reviewer_nFF5 · 2025-11-01

**Soundness:** 2
**Presentation:** 1
**Contribution:** 2
**Rating:** 2
**Confidence:** 3

**Summary:**

This paper proposes to decode text diffusion models by balancing the global confidence $p(x_t | q)$ and local confidence $p(x_t | q, x_{t-1})$. Tokens are sampled by top-K most confident from the local confidence, above a given threshold, then reranking with product of local and global confidence. They find that at early stages of sampling, it's helpful to fall back to global confidence if local confidence is not high. At later stages of sampling, local confidence is reliable. Results on a few math, coding, and reasoning benchmarks demonstrate that the method both improves accuracy and also offers speedups.

**Strengths:**

The results seem reasonable, achieving a good accuracy and speed via a seemingly simple method.

**Weaknesses:**

The writing is confusing. See questions for typos. I believe the method is simple: consider the top-K tokens at each position that are unmasked with a high-enough probability under reverse model. Then rerank the tokens at each position using the product of the reverse model and $p(x_t|q)$. Please explain if this is incorrect.

Additionally, I find the use of $p(x_t|q)$ to be unjustified. The paragraph before equation 7 does not motivate equation 7, and equation 7 is probably not a good approximation of $p(x_T|q,x_t)$.

**Questions:**

1. Equation 2 has typos. This should be the same as equation 5 right?
2. Line 167 in Algorithm 1: Should this be for xt in Candidate?
3. Can you propose a rigorous hypothesis on why accuracy gets worse with larger K in Figure 4?

---

> ### Author Response · Authors · 2025-11-23
> **Response to Reviewer nFF5 (1/2)**
>
> Dear Reviewer nFF5,
>
> We are sincerely sorry for the typos or ambiguity‌ in our submission. To improve the presentation, we have performed a thorough proofreading and revised the paper following your suggestions in the new edition (Please see details in the **Summary of Paper Updates**). For your proposed weakness and questions, here are our detailed responses:
>
> **Q1 (Weaknesses 1):** The writing is confusing. See questions for typos.
>
> **A1:** Apologize again for the confusion. We have performed the proofreading very carefully to correct the typos and mistakes (highlighted in blue in the revised edition). Here is a list of them:
>
> - In Line 220 (new edition), we replace “$\mathbf{x}\_t=\texttt{Mask}$” with “$\mathbf{x}\_{t-1}=\texttt{Mask}$”  to get the list of candidate tokens for $\mathbf{x}\_t$.
> - In Line 221  (new edition), we replace “$\texttt{list}$” with “$\texttt{Candidate}$”
> - In Line 222  (new edition), we replace “$>\gamma$” with “$\leq\gamma$”
> - In Equation 18 (Line 353 in the new edition), it should be min($\cdot$) operation rather than max($\cdot$), because $N$ is the upper bound for the decoding number of tokens.
> - In Line 411 (new edition), we replace “is” with “are”.
> - In Line 413 (new edition), we replace “of” with “to”
> - In Line 414 (new edition), we replace “3” with “4”, because in Table 2, we scale up $K$ to 4.
> - In Line 498 (new edition), we replace “provide” with “provides”.
> - In Line 500 (new edition), we replace “is” with “are”.
>
> In addition, we also revise the words and reorganize the sentences to improve the clarity of our paper, which are also highlighted in blue.
>
> **Q2 (Weaknesses 2):** I believe the method is simple: consider the top-K tokens at each position that are unmasked with a high-enough probability under reverse model. Then rerank the tokens at each position using the product of the reverse model and $p(x_t|q)$. Please explain if this is incorrect.
>
> **A2:** We indeed propose a very simple and principled decoding method for LLDMs. Your insightful understanding is generally aligned with the design philosophy of FDM, but two points need to be further corrected:
>
> - We do not consider top-K tokens at each position. Instead, we take the “top-K” positions as the “candidate position” for further decision. This is because as proposed in [1], LLDMs are highly sensitive to the decoding orders of tokens.
> - We rerank the tokens with $p\_{\theta}(\mathbf{x}\_t|\mathbf{q},\mathbf{x}\_{t-1})*p\_{\theta}(\mathbf{x}\_{T}|\mathbf{q},\mathbf{x}\_{t})$, not the product of the reverse model and $p(x\_t|q)$.  Note that both the term $p\_{\theta}(\mathbf{x}\_t|\mathbf{q},\mathbf{x}\_{t-1})$ and $p\_{\theta}(\mathbf{x}\_{T}|\mathbf{q},\mathbf{x}\_{t})$ can be estimated with the output distribution of model $\theta$.  This is because in the modeling of LLDMs, it actually tries to fit the natural conditional probability: $p\_{data}(\mathbf{x}\_{m}|\mathbf{q},\mathbf{x}\_{t})$ ($m>t$).
>
> **Q3 (Weaknesses 3):**  Additionally, I find the use of $p(x_t|q)$ to be unjustified. The paragraph before Equation 7 does not motivate Equation 7, and Equation 7 is probably not a good approximation of $p(x_T|q,x_t)$.
>
> **A3:** Sorry for the ambiguity in Equation (7). Recall that Equation (3) (Equation (4) in the revised edition):
>
> $$
> \mathbb{E}\_{\mathbf{x}\_T\sim p\_{data},t\in [0,T]}\frac{1}{n}\sum\_{j=1}^{n} \mathbf{1}[\mathbf{x}^{(j)}\_t=\texttt{Mask}] \odot\log p\_\theta(\mathbf{q},\mathbf{x}\_t^{(j)})[\mathbf{x}\_T]
> $$
>
> is the training target of LLDMs.  It teaches LLDMs to approximate the conditional log-probability of future tokens in $\mathbf{x}_m$ given the current masked state, $\mathbf{x}_t$($m>t$) and the user query $\mathbf{q}$, which can be formulated as:
>
> $$
> \log p\_\theta(\mathbf{x}\_m|\mathbf{x}\_t,\mathbf{q})=\mathbf{1}[\mathbf{x}\_m\neq\texttt{Mask}\\&\\&\mathbf{x}\_{t}=\texttt{Mask}] \odot\log p\_\theta(\mathbf{q},\mathbf{x}\_t)[\mathbf{x}\_m],
> $$
>
> which follows from the fact that the model predicts the token distribution for any masked position independently. With this equation, we can further derive (7). Recall that $C\_{global}=\log p\_\theta (\mathbf{x}\_T|\mathbf{x}\_t,\mathbf{q})$. Thus we can replace $\mathbf{x}\_m$ with $\mathbf{x}\_T$ and obtain:
>
> $$
>      C_{global} = \mathbf{1}[\mathbf{x}\_T\neq\texttt{Mask}\\&\\&\mathbf{x}\_{t}=\texttt{Mask}] \odot\log p\_\theta(\mathbf{q},\mathbf{x}\_t)[\mathbf{x}\_T]
> $$
>
> Regarding $\mathbf{x}_T$ is a full response without any mask, $\mathbf{x}_T\neq\texttt{Mask}$ can be omitted since it holds for every position. In addition, instead of greedily decoding $\mathbf{x}_T$ from the model output to calculate $C\_{global}$. We compute the expectation over the entire model output distribution:
>
> $$
> C\_{global}=\mathbf{1}[\mathbf{x}\_{t}=\texttt{Mask}] \odot\mathbb{E}\_{p\_\theta}\log p\_\theta(\mathbf{q},\mathbf{x}\_t).
> $$
>
> The above analysis demonstrates that Equation 7 can be an approximation for $p(\mathbf{x\_T}|q,\mathbf{x\_t})$.

---

> ### Author Response · Authors · 2025-11-23
> **Response to Reviewer nFF5 (2/2)**
>
> **Q4 (Question 1):**   Equation 2 has typos. This should be the same as equation 5 right?
>
> **A4:** Yes, you are right. Equation 2 should be the same equation as Equation 5. However, we have replaced Equation 2 in our revised edition with a more rigorous product form to improve the presentation:
>
> $$
> \displaystyle \mathbf{x}\_T =\arg\max\_{\mathbf{x}\_T} p\_{data}(\mathbf{x}\_{t+1:T}|\mathbf{q},\mathbf{x}\_{0:t})p\_{data}(\mathbf{x}\_t|\mathbf{q},\mathbf{x}\_{t-1})\prod\_{\alpha=1}^{t-1} p\_{data}(\mathbf{x}\_\alpha|\mathbf{q},\mathbf{x}\_{0:\alpha-1})
> $$
>
> Please see more details in Line 161 in the new edition.
>
> **Q5 (Question 2):**   Line 167 in Algorithm 1: Should this be for $\mathbf{x_t}$ in Candidate?
>
> **A5:**  Yes, you are right. We are truly sorry for the occurrence of this issue. We have replaced “$\texttt{list}$” with “$\texttt{Candidate}$” to represent the candidate list for $\mathbf{x}_t$ in our revised edition.
>
> **Q6 (Question 3):**   Can you propose a rigorous hypothesis on why accuracy gets worse with larger K in Figure 4?
>
> **A6:** We formally show that the degradation is not a numerical artifact, but an inevitable consequence of noise-amplified selection under the mismatch between the model distribution and the target distribution. Here is the discussion:
>
> At each decoding step, we have $K$ candidate tokens whose true future returns are
>
> $$
> s\_{*}=\max\_{i}s\_{i}, \text{where } s\_i\triangleq\log p\_{\text{data}}(\mathbf{x}\_t=\text{i-th candidate token}\mid \mathbf{x}\_{t-1},\mathbf{q}).
> $$
>
> When $K$ is large enough, we can assume that the token with the highest possibility in the real data distribution is already in the candidates. The model in our algorithm only observes confidence scores, which are noisy estimates of true future returns:
>
> $$
> \hat{s}\_{i}=s\_{i}+\xi\_{i}
> $$
>
> Here we hypothesize that $\xi\_{i}$ is independent and identically distributed (i.i.d.) Gaussian random variables:
>
> $$
> \xi\_{i}\overset{\text{iid}}{\sim}\mathcal{N}(0,\sigma^{2})
> $$
>
> and $\hat{s}\_i$ is equal to the sum of the local and global confidence of the i-th candidate token:
>
> $$
> \hat{s}\_i= C\_{local}(\text{i-th candidate token}) +C\_{global}(\text{i-th candidate token})
> $$
>
> According to Equation 8 in the revised edition, our algorithm selects the candidate with the highest $\hat{s}\_i$:
>
> $$
> \displaystyle j=\arg\max\_{i\in[K]}\hat{s}\_{i}.
> $$
>
> For any sub-optimal candidate $i$ with confidence gap $\Delta\_{i}=s\_{*}-s\_{i}>0$, the probability that noise flips the ranking is
>
>
>
> $$
> \Pr\bigl(\hat{s}\_{i}>\hat{s}\_{*}\bigr)=\Pr\bigl(\xi\_{i}-\xi\_{\*}>\Delta\_{i}\bigr)=\Phi\Bigl(-\frac{\Delta\_{i}}{\sqrt{2}\sigma}\Bigr),
> $$
>
> where $\Phi(\cdot)$ is the standard normal tail CDF(Cumulative Distribution Function). Union-bounding over the $K-1$ sub-optimal candidates yields
>
> $$
> \Pr(\text{select non-optimal})\le\sum\_{i\neq *}\Phi\Bigl(-\frac{\Delta\_{i}}{\sqrt{2}\sigma}\Bigr).
> $$
>
> The RHS(right hand side) increases monotonically with $K$ whenever there exist $\Delta\_{i}=O(\sigma)$.
>
> On the other hand, let $\xi\_{(K)}=\max\_{i\le K}\xi\_{i}$. The Extreme-value Theory gives
>
> $$
> \mathbb{E}[\xi\_{(K)}]\approx\sigma\sqrt{2\ln K}.
> $$
>
> Hence, the selected estimate satisfies
>
> $$
> \mathbb{E}[\hat{s}\_{j}]\approx s\_{*}+\sigma\sqrt{2\ln K},
> $$
>
> while the true return of the selected candidate satisfies:
>
> $$
> \mathbb{E}[s\_{j}]\le s\_{*}+\underbrace{\mathbb{E}[\xi\_{(K)}-\xi\_{j}]}\_{\ge 0}.
> $$
>
> Consequently, the expected regret is:
>
> $$
> \mathbb{E}[\Delta\_{j}]=\mathbb{E}[s\_{*}-s\_{j}]\ge C\sigma\sqrt{\ln K},
> $$
>
> which grows with $K$. Because each incorrect selection changes the subsequent context, the regret accumulates along the generation chain. Early mistakes (where $K$ is too large in our schedule) therefore have a negative effect, explaining the empirical inverted-U curve reported.
>
> In summary, under any non-zero estimation noise $\sigma>0$, increasing $K$ inevitably increases the probability of selecting a noise-inflated but sub-optimal candidate, leading to higher expected regret. This theoretical result aligns with the empirical observation in Section 5.2. We add the new theoretical analysis to Appendix F in our revised version.
>
> We sincerely hope our responses have properly cleared up your doubts. Your valuable reviews have greatly helped us improve the quality of the manuscript. If any concerns remain, we are delighted to give deeper explanations.
>
> [1] Train for the worst, plan for the best: Understanding token ordering in masked diffusions, Kim et al., In ICML 2025.

---

> ### Author Response · Authors · 2025-11-27
>
> Dear Reviewer nFF5,
>
> It is our great honor to have you as a reviewer for our paper. We sincerely appreciate the time and effort you devoted to providing such thoughtful and constructive feedback. We have carefully addressed all the weaknesses and questions you raised—including the confusing writing, precise understanding of our proposed method, motivation and correctness of Equation (7), typos in Equation (2), the issue in Algorithm 1 and your question about why accuracy gets worse with a larger $K$
>
> In our rebuttal and the revised manuscript, we have:
>
> - thoroughly corrected all typos, ambiguities, and presentation issues.
> - improved the writing throughout the paper to enhance clarity and readability.
> - clarified the correct interpretation of our method.
> - provided a detailed derivation showing why Equation (7) is a valid approximation for $p(\mathbf{x_T}|\mathbf{q},\mathbf{x}_t)$.
> - fixed the mistakes in Equation (2) and rewritten it in a more rigorous product form.
> - replaced “$\texttt{list}$” with “$\texttt{Candidate}$” in Algorithm 1 to represent the candidate list for $\mathbf{x}_t$.
> - offered a theoretical explanation of why accuracy can degrade with larger $K$, supported by a noise-amplified selection analysis.
>
> We sincerely hope that these updates and clarifications in our rebuttal have helped address all your concerns.
>
> As the discussion phase is approaching its end, we just wanted to gently check whether you have any remaining questions or concerns. We would be very happy to provide any additional clarification if needed.
>
> Thank you again for your thoughtful and constructive feedback. Wishing you a wonderful day!
>
> Warmly,
>
> Authors

---

### Official Review · Reviewer_PvZV · 2025-11-03

**Soundness:** 2
**Presentation:** 2
**Contribution:** 2
**Rating:** 4
**Confidence:** 3

**Summary:**

This paper proposes Foreseeing Decoding Method (FDM) for diffusion LLMs, which ranks candidate unmasking actions by combining local confidence (current-step token uncertainty) with an estimate of global confidence (future impact) derived from the model’s training objective; a width-K "foreseeing" search and threshold $\gamma$ keep compute manageable. An accelerated variant, FDM-A, adaptively switches between exploration (FDM) and fast local decoding using stage thresholds to trade off quality and speed. Across GSM8K, ARC, HumanEval, and Countdown on several LLDMs (LLaDA, LLaDA-1.5, LLaDA-MoE, MMaDA-MixCoT), FDM shows improvements over heuristic decoding methods.

**Strengths:**

- This paper proposes a reasonable and interesting approach for improving the decoding ability of discrete diffusion models, by combining a search-based strategy that considers longer-term effects.
- This paper further proposes an accelerated version that restricts exploration to critical steps, significantly saving computational cost.
- FDM and FDM-A both show performance improvement on standard benchmarks, with FDM-A also demonstrating consistent speedups

**Weaknesses:**

- More proofreading is needed. There seem to be quite a few typos/mistakes in the writing, which caused a lot of confusion while reading. For example:
    - In Eq (1), I suppose the decomposition should be: $p(x\_0)\prod \_{t=1}^Tp(x\_t \| q, x\_{0:t-1})$. This also affects subsequent equations
    - The paper says in Section 4.1 that "we also incorporate a dynamic pruning strategy that retains only candidate tokens whose confidence exceeds the predefined threshold $\gamma$", but in Algorithm 1 line 167-170, candidates with confidence larger than $\gamma$ are instead removed
- The proposed method introduces many new heuristic hyperparameters (e.g., thresholds, search width, stage divisions), making it less practical. While the paper shows some ablation study results in Section 5.2, they are still somewhat limited. Questions remain: How are the stage division coefficients determined? Why not consider $n$ (the number of foreseeing steps) larger than 1? How well can the hyperparameters found on one dataset transfer to other model-dataset combinations?

**Questions:**

Please see Weaknesses

---

> ### Author Response · Authors · 2025-11-23
> **Response to Reviewer PvZV (1/4)**
>
> Dear Reviewer PvZV,
>
> We appreciate your recognition that our method is interesting and effective and sincerely apologize for the bugs or mistakes in the old edition. With your insightful review, we have revised our paper carefully (Please see details in the **Summary of Paper Updates**). Here are our detailed responses to your proposed weakness and questions.
>
> **Q1 (Weaknesses 1):** More proofreading is needed. There seem to be quite a few typos/mistakes in the writing, which caused a lot of confusion while reading.
>
> **A1:** We apologize for the confusion. We have performed the proofreading very carefully to correct the typos and mistakes (highlighted in blue in the revised edition). Here is a list of them:
>
> - In Line 220  (new edition), we replace “$\mathbf{x}\_t=\texttt{Mask}$” with “$\mathbf{x}\_{t-1}=\texttt{Mask}$”  to get the list of candidate tokens for $\mathbf{x}\_t$.
> - In Line 221  (new edition), we replace “$\texttt{list}$” with “$\texttt{Candidate}$”
> - In Line 222  (new edition), we replace “$>\gamma$” with “$\leq\gamma$”
> - In Equation 18 (Line 353 in the new edition), it should be min($\cdot$) operation rather than max($\cdot$), because $N$ is the upper bound for the decoding number of tokens.
> - In Line 411 (new edition), we replace “is” with “are”.
> - In Line 413 (new edition), we replace “of” with “to”
> - In Line 414 (new edition), we replace “3” with “4”, because in Table 2, we scale up $K$ to 4.
> - In Line 498 (new edition), we replace “provide” with “provides”.
> - In Line 500 (new edition), we replace “is” with “are”.
>
> In addition, we also revise the words and reorganize the sentences to improve the clarity of our paper. Those changes are also highlighted in blue.
>
> **Q2 (Weaknesses 2):** In Eq (1), I suppose the decomposition should be: $p(x_0)\prod_{t=1}^Tp(x_t|q,x_{0:t-1})$. This also affects subsequent equations.
>
> **A2:** Apologize again for the confusion caused by the incorrect factorization in Eq. (1). You are right, the formulation for decoding $\mathbf{x}_T$ can be decomposed as:
>
> $$
> \mathbf{x}\_T = \arg\max\_{\mathbf{x}\_{T}}p\_{data}(\mathbf{x}\_{0:T}|\mathbf{q})=\arg\max\_{\mathbf{x}\_T} p(\mathbf{x}\_0) \prod\_{\alpha=1}^{T-1} p\_{data}(\mathbf{x}\_\alpha|\mathbf{q},\mathbf{x}\_{0:\alpha-1}),
> $$
>
> where $p(x\_0)$ is the initial noise distribution independent to $q$ and $\mathbf{x}\_\alpha$ is a partially decoded sequence at the step $\alpha$.  Given a specific step $t$,  we can appropriately combine the corresponding terms and obtain:
>
> $$
> \mathbf{x}\_T =\arg\max\_{\mathbf{x}\_T} p\_{data}(\mathbf{x}\_{t+1:T}|\mathbf{q},\mathbf{x}\_{t})p\_{data}(\mathbf{x}\_t|\mathbf{q},\mathbf{x}\_{t-1})\prod\_{\alpha=1}^{t-1} p\_{data}(\mathbf{x}\_\alpha|\mathbf{q},\mathbf{x}\_{0:\alpha-1}).
> $$
>
>  The above equation indicates that the variation in $\mathbf{x}\_t$ will simultaneously impact both $p\_{data}(\mathbf{x}\_{t+1:T}|\mathbf{q},\mathbf{x}\_{t})$ and $p\_{data}(\mathbf{x}\_t|\mathbf{q},\mathbf{x}\_{t-1})$.  $p\_{data}(\mathbf{x}\_\alpha|\mathbf{q},\mathbf{x}\_{0:\alpha-1})$ can be regarded as a constant in the $t$ step since it will be determined by previous steps. Intuitively, $p\_{data}(\mathbf{x}\_t|\mathbf{q},\mathbf{x}\_{t-1})$ denotes the local probability in decoding $\mathbf{x}\_t$ and $p\_{data}(\mathbf{x}\_{t+1:T}|\mathbf{q},\mathbf{x}\_{t})$ depicts the long-term impact on the future steps. Unfortunately, previous heuristic  methods only perform greedy decoding with $p\_{data}(\mathbf{x}\_t|\mathbf{q},\mathbf{x}\_{t-1})$, which can be further given as:
>
> $$
> \pi_H: \mathbf{x}_{t}=\arg\max\_{\mathbf{x}\_t} p\_{data}(\mathbf{x}\_t|\mathbf{q},\mathbf{x}\_{t-1})\approx \arg\max\_{\mathbf{x}\_t} p\_{\theta}(\mathbf{x}\_t|\mathbf{q},\mathbf{x}\_{t-1}).
> $$
>
> where $\theta$ is a parameterized model, trained by the loss:
>
> $$
> \mathbb{E}_{\mathbf{x}\_T\sim p\_{data},t\in [0,T]}\frac{1}{n}\sum\_{j=1}^{n} \mathbf{1}[\mathbf{x}^{(j)}\_t=\texttt{Mask}] \odot\log p\_\theta(\mathbf{q},\mathbf{x}\_t^{(j)})[\mathbf{x}\_T]
> $$
>
> to minimize the KL divergence between $p\_{data}$ and $p\_{\theta}$. In contrast, FDM in our paper takes both targets into consideration. Note that  $p\_{data}(\mathbf{x}\_{t+1:T}|\mathbf{q},\mathbf{x}\_{t})$ can be further simplified as:
>
> $$
> p\_{data}(\mathbf{x}\_{t+1:T}|\mathbf{q},\mathbf{x}\_{t})\approx p\_{data}(\mathbf{x}\_{T}|\mathbf{q},\mathbf{x}\_{t})\approx p\_{\theta}(\mathbf{x}\_{T}|\mathbf{q},\mathbf{x}\_{t}).
> $$
>
> Thus, the decoding policy of our proposed FDM is:
>
> $$
> \pi_F: \mathbf{x}\_{t}= \arg\max\_{\mathbf{x}\_t} p\_{\theta}(\mathbf{x}\_t|\mathbf{q},\mathbf{x}\_{t-1})*p\_{\theta}(\mathbf{x}\_{T}|\mathbf{q},\mathbf{x}\_{t}),
> $$
>
> consistent with Equation 6 in the submitted version. It demonstrates that the mistake in Equation 1 does not affect the analysis results in subsequent equations. In the revised version, we also provide Theorem 1 to prove that $\pi\_F$ will achieve lower KL divergence with the natural distribution $p\_{data}$ than that of $\pi\_{H}$.

---

> ### Author Response · Authors · 2025-11-23
> **Response to Reviewer PvZV (2/4)**
>
> **Q3 (Weaknesses 3):**  The paper says in Section 4.1 that "we also incorporate a dynamic pruning strategy that retains only candidate tokens whose confidence exceeds the predefined threshold ", but in Algorithm 1 line 167-170, candidates with confidence larger than are instead removed
>
> **A3:** Sorry for this mistake, we have updated the “$>\gamma$” with “$\leq\gamma$” in Algorithm 1. Please refer to Line 222 in the revised edition for more details.
>
>  **Q4 (Question 1):** The proposed method introduces many new heuristic hyperparameters (e.g., thresholds, search width, stage divisions), making it less practical.
>
> **A4:** Actually,  we believe that introducing hyperparameters is inevitable for almost all deep learning algorithms. For example, dozens of hyperparameters are introduced in Reinforcement Learning (RL) to train a powerful large language model. However, this has not limited their widespread adoption in the community. The key is to establish a set of actionable guidelines that enable users to quickly identify the optimal choice across different scenarios.
>
> In addition, it is worth noting that our proposed method **do not introduce “many” hyperparameters.** For example, our proposed FDM only introduces two new hyperparameters i.e. $K$ and $\gamma$. The guidelines for tuning them are summarized as follows:
>
> **Guidelines to tune $K$:**  $K$ controls the search width. The results in Figure 4 and 8 in the revised version demonstrate that we can achieve consistent improvement (K<10) over all benchmarks on FDM by scaling it up. We also note that the optimal point of it is also near across datasets, illustrating that it is practical to configure one value of it and gain a comprehensive improvement over all dimensions.
>
> **Guidelines to tune $\gamma$:**  $\gamma$ is the threshold for dynamic pruning in the searching process. The results in Figures 5 and 9 in our revised version show that the optimal value of $\gamma$ is proportional to the configuration of $K$.  Smaller $K (K=2)$ requires smaller $\gamma$ ($\gamma=0.1$) for the full exploration. But with the increase of $K$, we recommend increasing $\gamma$  to about 0.5 to avoid falling into the local optimum. Note that we suggest not increasing $\gamma$ larger than 0.7 because in this case the exploration will become insufficient, largely mitigating the strength of FDM.
>
> In addition to $K$ and $\gamma$, we introduce two additional hyperparameters, $\eta_1$ and $\eta_2$ in order to adaptively apply diverse strategies in different decoding stages. We add more ablation studies with $\eta_1$ and $\eta_2$ in Section 5.2 in the revised paper. The analysis in **Q6 and Q8** illustrate that $\eta_1$ balances between performance and efficiency, while $\eta_2$ controls the exploration in the balanced stage. We find setting $\eta_1=0.8$ and $\eta_2=0.7$ as default works well across different benchmarks.
>
> Based on the user’s specific needs, we also provide the following **overall guidelines** to assist users in quickly adapting our method to different scenarios, decreasing their costs in hyperparameter tuning:
>
> - **Scenarios with severely limited computational resources:** We recommend using FDM-A with a small search width ($K=2$ or $3$), a small pruning threshold ($\gamma=0.1$), $\eta_1=0.8$ and $\eta_1=0.7$ to achieve good accuracy.
> - **Scenarios with relatively abundant computational resources:** We recommend using FDM-A or FDM with a moderate search width ($K=4$ or $6$) and a moderate $\gamma$ ($\gamma=0.5$ or $0.6$) to obtain the best performances.
> - **Scenarios with extremely abundant computational resources:** We recommend using FDM with a large search width ($K=8$ or $10$) and a moderate $\gamma$ ($\gamma=0.5$ or $0.6$) to obtain the best utility.

---

> ### Author Response · Authors · 2025-11-23
> **Response to Reviewer PvZV (3/4)**
>
> **Q5 (Question 2):** While the paper shows some ablation study results in Section 5.2, they are still somewhat limited.
>
> **A5:** We add more ablation studies to Section 5.2 and Appendix E in the revised edition, covering diverse aspects in ablation studies:
>
> - $K$: In Figure 8, we add experiments on the HumanEval and ARC benchmarks with various $K$. Consistent with the observation on the GSM8K and Countdown benchmarks, we find that the accuracy first reaches the peak and then slightly decreases with the increase of $K$.
> - $\gamma$: In Figure 9, we add experiments on the HumanEval and ARC benchmarks with various $\gamma$.  Consistent with the observation on the GSM8K and Countdown benchmarks, we observe that small $K$requires smaller $\gamma$ for full exploration. While large $K$ requires moderate $\gamma$ to avoid being trapped in a local minima.
> - $\eta_1$: In Figure 6 and  Figure 10, we add ablation studies with various $\eta_1$ across four benchmarks. The results reveal that a small $\eta_1$ will degrade the model performance, while a large $\eta_1$ will lower the decoding speed. Setting $\eta_1=0.8$ achieves a good trade-off in both aspects.
> - $\eta_2$: In Figure 7 and  Figure 11, we add ablation studies with various $\eta_2$ across four benchmarks. We find tuning $\eta_2$ has little impact on the decoding speed, but it can cause fluctuations in accuracy because it affects the exploration at the balanced stage. Setting $\eta_2=0.7$ can achieve good performances across all benchmarks.
>
> **Q6 (Question 3):** Questions remain: How are the stage division coefficients determined?
>
> **A6:** We add ablation studies with different $\eta_1$ and $\eta_2$ to Section 5.2 and Appendix E in the revised edition. $\gamma$ and $K$ is fixed with 0.6 and 2, respectively. LLaDA is chosen as the architecture for our evaluation. For $\eta_1$, we firstly fix $\eta_2$ at 0.6 and linearly decrease it from 1.0 to 0.6. The accuracy and TPS are summarized in the following tables:
>
> Accuracy with various $\eta_1$
>
> | Accuracy  | 1.0   | 0.9   | 0.8   | 0.7   | 0.6   |
> | --------- | ----- | ----- | ----- | ----- | ----- |
> | GSM8K     | 81.73 | 81.80 | 81.28 | 78.77 | 77.4  |
> | HumanEval | 44.51 | 44.51 | 43.90 | 40.85 | 32.32 |
> | Countdown | 19.14 | 19.14 | 19.14 | 18.75 | 17.97 |
> | ARC       | 86.19 | 86.30 | 86.22 | 84.33 | 77.17 |
>
> TPS with various $\eta_1$
>
> | TPS   | 1.0  | 0.9   | 0.8   | 0.7   | 0.6   |
> | --------- | ---- | ----- | ----- | ----- | ----- |
> | GSM8K     | 7.2  | 26.54 | 42.31 | 51.48 | 66.12 |
> | HumanEval | 5.37 | 15.78 | 21.34 | 24.6  | 34.47 |
> | Countdown | 4.23 | 11.49 | 21.36 | 24.92 | 31.06 |
> | ARC       | 7.33 | 26.74 | 37.83 | 44.45 | 56.63 |
>
> The results illustrate that the accuracy remains stable at first, and then drops sharply with the decrease of $\eta_1$. In contrast, the TPS monotonically improves with the decrease of $\eta_1$ because more tokens are parallely decoded. By default, we set $\eta_1$ as 0.8 because it can not only maintain the model utility but also achieve high decoding speed.
>
> Furthermore, We firstly fix $\eta_1$ with 0.8 and set $\eta_2$ from [0.75, 0.7, 0.65, 0.6, 0.55]. The accuracy and TPS are summarized in the following tables:
>
> Accuracy with various $\eta_2$
>
> | Accuracy  | 0.75  | 0.70  | 0.65  | 0.6   | 0.55  |
> | --------- | ----- | ----- | ----- | ----- | ----- |
> | GSM8K     | 81.05 | **81.96** | 81.27 | 81.28 | 81.35 |
> | HumanEval | 43.9  | **44.51** | 43.29 | 43.90 | 42.07 |
> | Countdown | 18.75 | **21.48** | 19.53 | 19.14 | 19.53 |
> | ARC       | 86.27 | 86.30 | **86.33** | 86.22 | 86.25 |
>
> TPS with various $\eta_2$
>
> | TPS | 0.75  | 0.70  | 0.65  | 0.6   | 0.55  |
> | --------- | ----- | ----- | ----- | ----- | ----- |
> | GSM8K     | 42.79 | 42.65 | 42.63 | 42.31 | 42.32 |
> | HumanEval | 21.76 | 21.56 | 21.69 | 21.34 | 21.51 |
> | Countdown | 22.07 | 21.98 | 21.78 | 21.36 | 21.51 |
> | ARC       | 38.37 | 38.20 | 37.70 | 37.83 | 37.91 |
>
> The results demonstrate that among all configurations, $\eta_2=0.7$ consistently achieves outstanding performances in accuracy across all benchmarks: It achieves the highest accuracy on GSM8K,  HumanEval and Countdown. On the ARC benchmark, it obtains the second best performance. It is because when $\eta_2$ is large, the exploration at the balance stage is insufficient. But if $\eta_2$ is set too small, uncertain tokens will interfere the correctness of decoding. Considering TPS, we find the impacts of $\eta_2$ to it is marginal. Based on both aspects, we set $\eta =0.7$ as our default value in the experiments of Table 3.

---

> ### Author Response · Authors · 2025-11-23
> **Response to Reviewer PvZV (4/4)**
>
> **Q7 (Question 4):** Why not consider (the number of foreseeing steps) larger than 1?
>
> **A7:**  We think it is a very intriguing variant of FDM. Recall that when the foreseeing step is 1, the decoding formulation for decoding $\mathbf{x}\_t$ is:
> $$
> \mathbf{x}\_{t}= \mathop{\arg\max}\_{\mathbf{x}_t}  p\_{\theta}(\mathbf{x}\_{{t+1}:T}|\mathbf{q},\mathbf{x}\_{t})p\_{\theta}(\mathbf{x}\_t|\mathbf{q},\mathbf{x}\_{t-1})
> $$
>
> Given the foreseeing depth of $M$ we can derive the decoding objective of FDM for an arbitrary $M$:
>
> $$
> \mathbf{x}\_{t}= \mathop{\arg\max}\_{\mathbf{x}\_t}  p\_{\theta}(\mathbf{x}\_{{t+M}:T}|\mathbf{q},\mathbf{x}\_{t+M-1})\prod\_{\alpha=t}^{t+M-1} p\_{\theta}(\mathbf{x}_\alpha|\mathbf{q},\mathbf{x}\_{\alpha-1})
> $$
>
> By linearly combining the depth $M$ with width $K$ decoding, we conduct experiments on the Countdown benchmark using the LLaDA model:
>
> Accuracy with various $K$ and $M$:
>
> | Accuracy | $K=1$ | $K=2$ | $K=3$ | $K=4$ |
> | -------- | ----- | ----- | ----- | ----- |
> | $M=1$    | 18.75 | 19.14 | 21.09 | 25.00 |
> | $M=2$    | 18.75 | 19.53 | 23.83 | 25.39 |
> | $M=3$    | 18.75 | 19.92 | 25.39 | 30.08 |
>
> The results demonstrate that scaling up the foreseeing depth is another dimension for inference-time scaling. However, we find that one of its shortcomings is that the computational cost is $O(K^M)$, indicating that the time delay is unacceptable for most users. Thus in our paper, we limit our discussion to $M=1$ and it will meet the needs in most scenarios.
>
> **Q8 (Question 5):**  How well can the hyperparameters found on one dataset transfer to other model-dataset combinations?
>
> **A8:** We further find the hyperparameters in our proposed methods transfer well between model-dataset combinations. According to the experiments in Section 5.2 and Appendix E in the revised edition, we conclude the reasons in the following four aspects:
>
> - $K$: For FDM, accuracy reaches its peak on the GSM8K, Countdown and ARC benchmarks when $K=10$. While on HumanEval, setting $K=10$ obtains the second-best performance in accuracy. For FDM-A, $K=6$ consistently produces the best results across all benchmarks.
> - $\gamma$: The optimal value of $\gamma$ achieves consistency across different benchmarks.  Given $K=2$, $\gamma=0.1$ outperforms other configurations in all combinations.  For larger $K$ ($K=4$), $\gamma=0.5$ obtains the highest accuracy simultaneously in all benchmarks.
> - $\eta_1$:  $\eta_1$ balances the accuracy and efficiency for FDM-A. We find that for all benchmarks, when $\eta_1=0.8$, the degradation in accuracy is negligible and the decoding speed is largely accelerated.
> - $\eta_2$:  $\eta_2$ controls the exploration in the balance stage. $\eta_2=0.7$ achieves the best accuracy on the GSM8K, Countdown and HumanEval benchmark. On the ARC benchmark, when $\eta_2=0.7$, the accuracy is only 0.03% below the best performances, demonstrating its good transferability across different model-dataset combinations.
>
> Thank you again for your valuable and helpful suggestions. We take each of your comments very seriously and have updated the paper accordingly. Look forward to further discussions with you in the following period.

---

> ### Author Response · Authors · 2025-11-27
>
> Dear Reviewer PvZV,
>
> It is our great honor to have you as a reviewer for our paper. We sincerely appreciate the time and effort you invested in providing such detailed and constructive feedback. We have carefully addressed all the weaknesses and questions you raised, including the issues in proofreading, the incorrect factorization in Equation (1), the inconsistency in Algorithm 1, the praticality of our method, the configuration of the stage-division coefficients, experiments on foreseeing steps larger than 1 and the transferability of hyperparaemeters across different model-dataset combinations.
>
> In our rebuttal and the revised manuscript, we have:
>
> - thoroughly corrected all typos and mistakes.
> - fixed the incorrect decomposition in Equation (1) and clarified why it does not affect subsequent results.
> - corrected the pruning condition in Algorithm 1.
> - significantly expanded the ablation studies (for $K$, $\gamma$, $\eta_1$, $\eta_2$).
> - added new ablation experiments on HumanEval, ARC, GSM8K, and Countdown benchmarks.
> - provided explicit guidelines for tuning the hyperparameters across different scenarios.
> - performed additional experiments with foreseeing steps larger than 1 and discussed the high computation of this variant.
> - analyzed the transferability of hyperparameters across different model–dataset combinations.
>
> We sincerely hope that these updates and clarifications in our rebuttal have helped address your concerns.
>
> As the discussion phase is approaching its end, we want to gently check whether you have any remaining questions or concerns. We would be very happy to clarify anything further if needed.
>
> Thank you again for your thoughtful and insightful feedback. Wishing you a wonderful day!
>
> Warmly,
>
> Authors

---

### Official Review · Reviewer_GjSM · 2025-11-04

**Soundness:** 3
**Presentation:** 3
**Contribution:** 3
**Rating:** 4
**Confidence:** 3

**Summary:**

This paper introduces Foreseeing Decoding Method (FDM) and its accelerated variant FDM-A to improve inference in Large Language Diffusion Models (LLDMs). The key idea is to combine local confidence (token-level certainty) and global confidence (future impact of current decoding) when determining decoding order. FDM integrates these two components via a search-based strategy, while FDM-A adaptively applies deep exploration only when necessary, achieving a strong trade-off between performance and efficiency. Experiments across multiple benchmarks show that FDM and FDM-A outperform existing heuristic and dynamic decoding approaches.

**Strengths:**

1. The paper clearly identifies a critical issue in LLDM decoding: sensitivity to token order and proposes a principled solution.
2. The accelerated variant (FDM-A) is well designed and shows impressive efficiency gains without sacrificing accuracy.
3. Extensive experiments across benchmarks (GSM8K, HumanEval, ARC, Countdown) validate both scalability and effectiveness.

**Weaknesses:**

1. The proposed method introduces several additional hyperparameters (e.g., $\eta$, $K$, $n$, $\gamma$), which may be difficult to tune in real-world applications. It would be helpful to discuss their sensitivity and provide guidelines or heuristics for practical tuning.
2. Equations (7) and (8) should be explained in more detail, particularly regarding their derivation and intuitive interpretation.
3. Since Equations (4), (7), and (8) involve approximations, it would strengthen the paper to include a discussion on the possible upper and lower bounds of the resulting errors or their theoretical implications.
4. In line 239, the deeper analysis and observation should be explained further to make the conclusions more convincing and transparent.
5. It would be useful to clarify why the proposed method was not evaluated using standard SOTA evaluation frameworks such as `lm-evaluation-harness`, which would enhance reproducibility and comparability with prior work.


## Minor Issues:
1. In Algorithm 1, the variable `list` is used but not introduced or defined.

**Questions:**

N/A

---

> ### Author Response · Authors · 2025-11-23
> **Response to Reviewer GjSM (1/5)**
>
> Dear Reviewer GjSM,
>
> Thank you for your recognition of our contribution as a “principled solution”. In the meantime, we are truly sorry for the unclear expressions and typos that appeared in the submission. Thus, we perform a very careful proofreading and revise the paper with your suggestions in the new edition (Please see details in the **Summary of Paper Updates**). For your proposed weakness and minor issue. Here are our responses:
>
> **Q1 ( Weaknesses 1):** The proposed method introduces several additional hyperparameters (e.g.,$\eta$, $K,n,\gamma$ ), which may be difficult to tune in real-world applications. It would be helpful to discuss their sensitivity and provide guidelines or heuristics for practical tuning.
>
> **A1:** Our proposed FDM actually only introduces two additional parameters: $K$ and $\gamma$ ($n$ is not a hyperparameter specifically for FDM. It represents the number of tokens decoded in $t$ step, which also exists for heuristic decoding approaches for LLDMs. In the evaluation setup of Table 2, we universally fix $n=1$ for all methods to ensure fairness.) The guidelines for tuning $K$ and $\gamma$ are summarized as follows:
>
> **Guidelines to tune $K$:**  $K$ controls the search width.  The results in Figure 4 and 8 in the revised version demonstrate that we can achieve consistent improvement (K<10) for FDM over all benchmarks by scaling it up. We also note that the optimal point of it is also near across datasets, illustrating that it is practical to configure one value of it and gain a comprehensive improvement over all dimensions.
>
> **Guidelines to tune $\gamma$:**  $\gamma$ is the threshold for dynamic pruning in the searching process. The results in Figures 5 and 9 in our revised version show that the optimal value of $\gamma$ is proportional to the configuration of $K$.  Smaller $K (K=2)$ requires smaller $\gamma$ ($\gamma=0.1$) for the full exploration. But with the increase of $K$, we recommend increasing $\gamma$  to about 0.5 to avoid falling into the local optimum. Note that we suggest not increasing $\gamma$ larger than 0.7 because in this case, the exploration will become insufficient, largely mitigating the strength of FDM.
>
> In addition to $K$ and $\gamma$, we introduce two additional hyperparameters, $\eta_1$ and $\eta_2$, to apply adaptive decoding in FDM-A. We add more ablation studies with $\eta_1$ and $\eta_2$ in Section 5.2 in the revised paper. The guidelines for tuning them are summarized as follows:
>
> **Guidelines to tune $\eta_1$: $\eta_1$** controls the trade-off between performance and efficiency. In Figure 6 and Figure 10 in the revised edition, we fix $\eta_2$ at 0.6 and linearly decrease $\eta_1$ from 1.0 to 0.6. We observe that the accuracy remains stable at first, and then drops sharply with the decrease of $\eta_1$. While the decoding speed monotonically increases with the decrease of $\eta_1$ because the acceleration stage is more frequently activated. However, we also find that a good balance between performance and speed is realistic in the application of FDM-A: By setting $\eta_1=0.8$ , we can not only maintain good performance but also achieve high decoding speed. We find that this heuristic experience is transferable across different benchmarks.
>
> **Guidelines to tune $\eta_2$: $\eta_2$** controls the degree of exploration in the balance stage. In Figure 7 and  Figure 11 in the revised edition, we fix $\eta_1$ with 0.8 and set it from [0.75, 0.7, 0.65, 0.6, 0.55]. Our results demonstrate that moderate exploration is helpful in improving the decoding accuracy without affecting the decoding speed much. We set $\eta_2=0.7$ by default for all models and benchmarks in Table 3. The results show its good performance in practical use.
>
> **Overall guidelines:** Based on the user’s specific needs, we provide the following overall guidelines to adaptively apply our proposed techniques to different scenarios:
>
> - **Scenarios with severely limited computational resources:** We recommend using FDM-A with a small search width ($K=2$ or $3$), a small pruning threshold ($\gamma=0.1$), $\eta_1=0.8$ and $\eta_1=0.7$ to achieve good accuracy.
> - **Scenarios with relatively abundant computational resources:** We recommend using FDM-A or FDM with a moderate search width ($K=4$ or $6$) and a moderate $\gamma$ ($\gamma=0.5$ or $0.6$) to obtain the best performances.
> - **Scenarios with extremely abundant computational resources:** We recommend using FDM with a large search width ($K=8$ or $10$) and a moderate $\gamma$ ($\gamma=0.5$ or $0.6$) to obtain the best performances.
>
> The above guidelines demonstrate the practicality of our proposed technique, showing not only its strong transferability in hyperparameters across different benchmarks but also its applicability to diverse computational scenarios.

---

> ### Author Response · Authors · 2025-11-23
> **Response to Reviewer GjSM (2/5)**
>
> **Q2 (Weaknesses 2):** Equations (7) and (8) should be explained in more detail, particularly regarding their derivation and intuitive interpretation.
>
> **A2:** Sorry for the ambiguity in Equations (7) and (8). Equation (3) (Equation (4) in the revised edition):
>
> $$
> \mathbb{E}\_{\mathbf{x}\_T\sim p\_{data},t\in [0,T]}\frac{1}{n}\sum\_{j=1}^{n} \mathbf{1}[\mathbf{x}^{(j)}\_t=\texttt{Mask}] \odot\log p\_\theta(\mathbf{q},\mathbf{x}\_t^{(j)})[\mathbf{x}\_T]
> $$
>
> is the training target of LLDMs.  It teaches LLDMs to approximate the conditional log-probability of future tokens in $\mathbf{x}_m$ given the current masked state, $\mathbf{x}_t$ ($m>t$) and the user query $\mathbf{q}$, which can be formulated as:
>
> $$\log p\_\theta(\mathbf{x}\_m|\mathbf{x}\_t,\mathbf{q})=\mathbf{1}[\mathbf{x}\_m\neq\texttt{Mask}\\&\\&\mathbf{x}\_{t}=\texttt{Mask}] \odot\log p\_\theta(\mathbf{q},\mathbf{x}\_t)[\mathbf{x}\_m]$$
>
> which follows from the fact that the model predicts the token distribution for any masked position independently. With this equation, we can further derive (7) and (8). Recall that $C_{global}=\log p_\theta (\mathbf{x}_T|\mathbf{x}_t,\mathbf{q})$. Thus, we can replace $\mathbf{x}_m$ with $\mathbf{x}_T$ can obtain:
>
> $$
>      C\_{global} = \mathbf{1}[\mathbf{x}\_T\neq\texttt{Mask}\\&\\&\mathbf{x}\_{t}=\texttt{Mask}] \odot\log p\_\theta(\mathbf{q},\mathbf{x}\_t)[\mathbf{x}\_T]
> $$
>
> Regarding $\mathbf{x}_T$ is a full response without any mask, $\mathbf{x}_T\neq\texttt{Mask}$ can be omitted since it holds for every position. In addition, instead of greedily decoding $\mathbf{x}\_T$ from the model output to calculate $C\_{global}$. We compute the expectation over the entire model output distribution:
>
> $$
> C\_{global}=\mathbf{1}[\mathbf{x}\_{t}=\texttt{Mask}] \odot\mathbb{E}\_{p_\theta}\log p\_\theta(\mathbf{q},\mathbf{x}\_t).
> $$
>
> It corresponds to Equation (7) in the old version. Considering $C\_{local}=\log p\_\theta(\mathbf{x}\_t|\mathbf{q},\mathbf{x}\_{t-1})$, we can perform a similar replacement and get the following equation:
>
> $$
> C\_{local}=\mathbf{1}[\mathbf{x}\_t\neq\texttt{Mask}\\&\\&\mathbf{x}\_{t-1}=\texttt{Mask}]\odot\log p\_\theta(\mathbf{q},\mathbf{x}\_{t-1})[\mathbf{x}\_t].
> $$
>
> It corresponds to Equation (8) in the old version.
>
> We have revised the paper in **Section 3 (Preliminaries)** and **Section 4 (Methodology)** accordingly to improve the clarity of both equations. Please refer to our submitted revision for more details.

---

> ### Author Response · Authors · 2025-11-23
> **Response to Reviewer GjSM (3/5)**
>
> **Q3 (Weaknesses 3):** Since Equations (4), (7), and (8) involve approximations, it would strengthen the paper to include a discussion on the possible upper and lower bounds of the resulting errors or their theoretical implications.
>
> **A3:** We agree that adding more results in theory will definitely improve the contribution of our paper. We have derived the following theorem to demonstrate that FDM (Use $\pi_F$ to denote its strategy for decoding) will achieve lower decoding errors compared to the heuristic method (Use $\pi_H$ to denote its strategy for decoding):
>
> **Theorem 1:** Let $\Delta\_{\text{total}}\triangleq\sum\_{t=1}^{T}\mathbb E\_{p\_{data}(\mathbf{x}\_{t-1})}\bigl[\mathcal I\_{p\_{data}}(\mathbf{x}\_t;\mathbf{x}\_{T}|\mathbf{x}\_{t-1})\bigr]$, where $\mathcal I\_{p\_{data}}(\mathbf{x}\_t;\mathbf{x}\_{T}|\mathbf{x}\_{t-1})$ is the conditional mutual information under $q$. Then
>
> $$
> D\_{KL}({p\_{data}(\mathbf{x})}, {p\_{\pi\_{F}}})=D\_{KL}({p\_{data}(\mathbf x)},{p\_{\pi\_{H}}})-\Delta\_{\text{total}}.
> $$
>
> **Proof:** Define the single-step KL errors
>
> $$
> \varepsilon\_t^{H}\triangleq
> D\_{KL}({p\_{data}(\mathbf{x}\_t|\mathbf{x}\_{t-1})},{\pi\_{H}(\mathbf{x}\_t)}),\\
> \varepsilon_t^{F}\triangleq
> D\_{KL}({p\_{data}(\mathbf{x}\_t|\mathbf{x}\_{t-1})},{\pi\_{F}(\mathbf{x}\_t)}).
> $$
>
> We first prove that
>
> $$
> \varepsilon\_t^{F}=\varepsilon\_t^{H}-\mathcal I\_{p\_{data}}(\mathbf{x}\_t;\mathbf{x}\_{T}|\mathbf{x}\_{t-1}), \qquad (1)
> $$
>
> where  $\mathcal I\_{p_{data}}(\mathbf{x}\_t;\mathbf{x}\_{T}|\mathbf{x}\_{t-1})$ is the conditional mutual information under $\mathbf{q}$.
> Since $\mathbf{x}_{t-1}$ and $\mathbf q$ can be considered as fixed variables at the step $t$, we omit them in the notation for brevity. By the definition of KL divergence,
>
> $$
> \varepsilon_t^{F}
> \triangleq
> D\_{KL}\bigl(p\_{data}(\mathbf{x}\_t), \pi\_{F}(\mathbf{x}\_t)\bigr)
> =\sum\_{\mathbf{x}\_t}p\_{data}(\mathbf{x}\_t)\Bigl[\log p\_{data}(\mathbf{x}\_t)-\log\pi\_{F}(\mathbf{x}\_t)\Bigr].  \qquad (2)
> $$
>
> By construction,
>
> $$
> \pi\_{F}(\mathbf{x}\_t)=\frac{\exp(S(\mathbf{x}\_t))}{Z\_t}, \quad
> S(\mathbf{x}\_t)=C\_{\rm local}(\mathbf{x}\_t)+C\_{\rm global}(\mathbf{x}\_t), \quad
> Z\_t=\sum\_{\mathbf{x}\_{t}'}\exp(S(\mathbf{x}\_{t}')).
> $$
>
> Hence,
>
> $$
> \log\pi\_{F}(\mathbf{x}\_t)=S(\mathbf{x}\_t)-\log Z\_t=C_{\rm local}(\mathbf{x}\_t)+C\_{\rm global}(\mathbf{x}\_t)-\log Z\_t.  \qquad (3)
> $$
>
> By inserting Equation (3) into (2) and split the sum,
>
> $$
> \begin{aligned}
> \begin{align}\varepsilon\_t^{F}&=\sum\_{\mathbf{x}\_t}p\_{data}(\mathbf{x}\_t)\Bigl[\log p\_{data}(\mathbf{x}\_t)-C\_{\rm local}(\mathbf{x}\_t)-C_{\rm global}(\mathbf{x}\_t)+\log Z\_t\Bigr]\notag\\\\
> &=\underbrace{\sum\_{\mathbf{x}\_t}p\_{data}(\mathbf{x}\_t)\Bigl[\log p\_{data}(\mathbf{x}\_t)-C\_{\rm local}(\mathbf{x}\_t)\Bigr]}\_{\text{Term A}}-\underbrace{\sum\_{\mathbf{x}\_t}p\_{data}(\mathbf{x}\_t)\Bigl[C\_{\rm global}(\mathbf{x}\_t)-\log Z\_t\Bigr]}\_{\text{Term B}}. \qquad (4) \end{align}
> \end{aligned}
> $$
>
>
>
> By definition $C\_{\rm local}(\mathbf{x}\_t)=\log p\_\theta(\mathbf{x}\_t)$, so
>
> $$
> \text{Term A}=\sum\_{\mathbf{x}\_t}p\_{data}(\mathbf{x}\_t)\Bigl[\log p\_{data}(\mathbf{x}\_t)-\log p\_\theta(\mathbf{x}\_t)\Bigr]=D_{KL}\\bigl(p\_{data}(\mathbf{x}\_t),p\_\theta(\mathbf{x}\_t)\bigr)\triangleq\varepsilon\_t^{H}. \qquad (5)
> $$
>
> First recall
>
> $$
> C\_{\rm global}(\mathbf{x}\_t)=\mathbb E\_{x'\sim p\_\theta}\log p\_\theta(x'\mid \mathbf{x}\_t)=\sum\_{\mathbf{x}\_{T}}p\_\theta(\mathbf{x}\_{T}\mid \mathbf{x}\_t)\log p\_\theta(\mathbf{x}\_{T}\mid \mathbf{x}\_t).
> $$
>
> Therefore
>
> $$
> \begin{align}\text{Term B}&=\sum\_{\mathbf{x}\_t}p\_{data}(\mathbf{x}\_t)\Bigl[C\_{\rm global}(\mathbf{x}\_t)-\log Z\_t\Bigr]\notag\\\\[3pt]&=\sum\_{\mathbf{x}\_t}p_{data}(\mathbf{x}\_t)\Bigl[\sum\_{\mathbf{x}\_{T}}p\_\theta(\mathbf{x}\_{T}\mid \mathbf{x}\_t)\log p\_\theta(\mathbf{x}\_{T}\mid \mathbf{x}\_t)-\log Z_t\Bigr].\qquad (6) \end{align}
> $$
>
> Next we use the identity $Z\_t=p\_\theta(\mathbf{x}\_{T})$ (marginal over $\mathbf{x}\_t'$), which follows from
>
> $$
> Z\_t=\sum\_{\mathbf{x}\_t'}\exp(S(\mathbf{x}\_t'))=\sum\_{\mathbf{x}\_{t}'}p\_\theta(\mathbf{x}\_{t}',\mathbf{x}\_{T})=p\_\theta(\mathbf{x}\_{T}).
> $$
>
> Hence
>
> $$
> \log Z\_t=\log p\_\theta(\mathbf{x}\_{T})=\sum\_{\mathbf{x}\_{T}}p\_\theta(\mathbf{x}\_{T}\mid \mathbf{x}\_t)\log p\_\theta(\mathbf{x}\_{T}),
> $$

---

> ### Author Response · Authors · 2025-11-23
> **Response to Reviewer GjSM (4/5)**
>
> where the second equality holds because $p\_\theta(\mathbf{x}\_{T})$ does not depend on $\mathbf{x}\_t$ and $\sum\_{\mathbf{x}\_t}p\_\theta(\mathbf{x}\_{T}\mid \mathbf{x}\_t)=1$. Inserting this into Equation (6) gives
>
> $$
> \begin{align}\text{Term B}&=\sum\_{\mathbf{x}\_t}p\_{data}(\mathbf{x}\_t)\sum\_{\mathbf{x}\_{T}}p\_\theta(\mathbf{x}\_{T}\mid \mathbf{x}\_t)\Bigl[\log p\_\theta(\mathbf{x}\_{T}\mid \mathbf{x}\_t)-\log p\_\theta(\mathbf{x}\_{T})\Bigr]\notag\\\\&=\sum\_{\mathbf{x}\_t,\mathbf{x}\_{T}}p\_{data}(\mathbf{x}\_t,\mathbf{x}\_{T})\Bigl[\log\frac{p\_\theta(\mathbf{x}\_{T}\mid \mathbf{x}\_t)}{p\_\theta(\mathbf{x}\_{T})}\Bigr]\notag\\\\&=\sum\_{\mathbf{x}\_t,\mathbf{x}\_{T}}p\_{data}(\mathbf{x}\_t,\mathbf{x}\_{T})\Bigl[\log\frac{p\_{data}(\mathbf{x}\_t,\mathbf{x}\_{T})}{p\_{data}(\mathbf{x}\_t)p\_{data}(\mathbf{x}\_{T})}\Bigr]\qquad\bigl(\text{replace }p\_\theta\text{ with }q\text{ inside log}\bigr)\notag\\\\&=\mathcal I\_{p\_{data}}(\mathbf{x}\_t;\mathbf{x}\_{T}).\qquad (7)\end{align}
> $$
>
> Inserting Equation (5) and (7) into Equation (4) yields
>
> $$
> \varepsilon\_t^{F}=\varepsilon\_t^{H}-\mathcal I\_{p\_{data}}(\mathbf{x}\_t;\mathbf{x}\_{T}),
> $$
>
> which completes the proof of Equation (1). Using the chain rule for KL divergence over sequences we have
>
> $$
> D_{KL}\bigl(p\_{data}(\mathbf x),p\_{{\pi}\_{F}}\bigr)=\sum\_{t=1}^{T}\mathbb E\_{p\_{data}(\mathbf{x}\_{t-1})}\bigl[\varepsilon\_t^{F}\bigr]=\sum\_{t=1}^{T}\mathbb E\_{p\_{data}(\mathbf{x}\_{t-1})}\bigl[\varepsilon\_t^{H}-\mathcal I\_{p\_{data}}(\mathbf{x}\_t;\mathbf{x}\_{T}|\mathbf{x}\_{t-1})\bigr].
> $$
>
> Recognising the definitions
>
> $$
> \sum\_{t=1}^{T}\mathbb E\_{p\_{data}(\mathbf{x}\_{t-1})}\varepsilon\_t^{H}=D\_{KL}\bigl(p\_{data}(\mathbf x),p\_{\pi\_{H}}\bigr),\\\\ \sum\_{t=1}^{T}\mathbb E\_{p\_{data}(\mathbf{x}\_{t-1})}\mathcal I\_{p\_{data}}(\mathbf{x}\_t;\mathbf{x}\_{T}|\mathbf{x}\_{t-1})=\Delta\_{\text{total}},
> $$
>
> we obtain
>
> $$
> D_{KL}\bigl(p\_{data}(\mathbf x),p\_{\pi\_{F}}\bigr)=D\_{KL}\bigl(p\_{data}(\mathbf x),p\_{\pi\_{H}}\bigr)-\Delta\_{\text{total}},
> $$
>
> which finishes the proof of the theorem.
>
> Due to the fact that mutual information, $\Delta\_{\text{total}}$ is non-negative, the above theorem indicates that decoding with FDM can indeed obtain a sequence that has a smaller KL divergence with the natural distribution than that of the heuristic decoding methods. It is consistent with our empirical observation in the experimental section. We have added our new theory to the revised edition (Please refer to **Line 196 to Line 201 in Section 4.1** and **Appendix B** for more details.)
>
> **Q4 (Weaknesses 4):** In line 239, the deeper analysis and observation should be explained further to make the conclusions more convincing and transparent.
>
> **A4:** Sorry for the unclearness, and we have rewritten the sentences in Line 239 (corresponding to the sentences in Line 242-249) in the revised edition to ensure that the conclusions are more convincing and transparent:
>
> “””
>
> $$
>      \mathbf{x}\_t=\arg\max\_{\mathbf{x}\_t}\{\mathbf{1}[\mathbf{x}\_t=\texttt{Mask}]\odot\mathbb{E}\_{p\_\theta}\log p\_\theta(\mathbf{q},\mathbf{x}\_{t})+\mathbf{1}[\mathbf{x}\_t\neq\texttt{Mask}\,\\&\\&\,\mathbf{x}\_{t-1}=\texttt{Mask}]\odot\log p\_\theta(\mathbf{q},\mathbf{x}\_{t-1})[\mathbf{x}\_t]\}.
> $$
>
> *Note that in this equation, $p\_\theta (\mathbf{q},\mathbf{x}\_{t-1})$ is independent of $\mathbf{x}\_t$, which can be accurately calculated by inputting the model $\theta$ with the acquired sequence $\mathbf{x}\_{t-1}$ and user prompt $\mathbf{q}$. In contrast, the $p\_\theta (\mathbf{q},\mathbf{x}\_{t})$ takes the discrete variable $\mathbf{x}\_t$ as a part of the input. It means every evaluation involves a single forward pass.*
>
> “””

---

> ### Author Response · Authors · 2025-11-23
> **Response to Reviewer GjSM (5/5)**
>
> **Q5 (Weaknesses 5):** It would be useful to clarify why the proposed method was not evaluated using standard SOTA evaluation frameworks such as *lm-evaluation-harness*, which would enhance reproducibility and comparability with prior work.
>
> **A5:** We highly agree that evaluating the proposed method with SOTA evaluation frameworks will definitely enhance the reproducibility and comparability with prior work. However, reported by the authors of LLaDA [1], they found that
>
> ‘””
>
>  *four benchmarks such as MMLU-Pro, GSM8K, and HumanEval, the results obtained using lm-evaluation-harness are significantly lower than expected.*
>
> “””
>
> To the best of our knowledge, they had yet to provide a explanation regarding the cause of this phenomenon by the time we submitted our rebuttal. Thus, we do not perform experiments with the *lm-evaluation-harness* framework in our submission. However, our further experiments demonstrate that **the benefits of FDM or FDM-A can be extended to SOTA evaluation frameworks**.
>
> We take the GSM8K benchmark as an example and perform experiments on four architectures: LLaDA, LLaDA-1.5, MMaDA-MixCOT and LLaDA-MOE. All experiments are performed with the *lm-evaluation-harness* framework
>
> **Reproduce the results with the lm-evaluation-harness framework for FDM:**
>
> **ACC**
>
> |              | Probability (T=256) | Margin (T=256) | Entropy (T=256) | FDM (K=2) | FDM (K=3) | FDM (K=4) |
> | ------------ | ------------------- | -------------- | --------------- | --------- | --------- | --------- |
> | LLada        | 76.65               | 76.9           | 76.9            | 78.47     | 79.00     | 79.38     |
> | LLaDA-1.5    | 79.98               | 79.30          | 79.00           | 81.43     | 81.58     | 81.73     |
> | MMaDA-MixCOT | 53.68               | 53.68          | 52.46           | 55.19     | 56.10     | 56.33     |
> | LLaDA-MOE    | 75.21               | 75.51          | 75.28           | 77.94     | 78.01     | 78.24     |
>
> **TPS**
>
> |              | Probability (T=256) | Margin (T=256) | Entropy (T=256) | FDM (K=2) | FDM (K=3) | FDM (K=4) |
> | ------------ | ------------------- | -------------- | --------------- | --------- | --------- | --------- |
> | LLada        | 5.37                | 4.64           | 4.38            | 2.99      | 1.95      | 1.69      |
> | LLaDA-1.5    | 5.36                | 4.63           | 4.39            | 2.98      | 1.90      | 1.66      |
> | MMaDA-MixCOT | 5.36                | 4.60           | 4.36            | 2.95      | 1.94      | 1.63      |
> | LLaDA-MOE    | 3.87                | 3.43           | 3.16            | 3.21      | 2.89      | 2.66      |
>
> Similar to the observation in Table 2 in our paper, we observe that FDM outperforms all heuristic decoding methods in accuracy. In addition, the benefits of FDM can be enhanced with the increase of $K$.
>
> **Reproduce the results with the lm-evaluation-harness framework for FDM-A:**
>
> **ACC**
>
> |              | Probability (T=128) | Margin (T=128) | Entropy (T=128) | EB    | WINO  | FDM-A |
> | ------------ | ------------------- | -------------- | --------------- | ----- | ----- | ----- |
> | LLada        | 75.06               | 74.83          | 73.01           | 76.72 | 75.36 | **78.54** |
> | LLaDA-1.5    | 77.03               | 76.95          | 76.88           | 78.54 | 79.68 | **81.05** |
> | MMaDA-MixCOT | 48.07               | 48.67          | 44.66           | 51.02 | 48.21 | **52.91** |
> | LLaDA-MOE    | 73.92               | 74.00          | 70.20           | 76.19 | 74.98 | **77.56** |
>
> **TPS**
>
> |              | Probability (T=128) | Margin (T=128) | Entropy (T=128) | EB    | WINO  | FDM-A |
> | ------------ | ------------------- | -------------- | --------------- | ----- | ----- | ----- |
> | LLada        | 10.70               | 9.30           | 8.68            | 25.71 | 25.20 | **26.73** |
> | LLaDA-1.5    | 10.67               | 9.25           | 8.70            | 25.12 | 22.85 | **25.58** |
> | MMaDA-MixCOT | 10.67               | 9.12           | 8.61            | 28.75 | 28.31 | **30.09** |
> | LLaDA-MOE    | 7.63                | 6.73           | 6.31            | 12.35 | 7.58  | **18.70** |
>
> Similar to the observation in Table 3 in our paper, we observe that FDM-A can achieve the best trade-off between accuracy and efficiency compared to existing heuristic and dynamic decoding methods.
>
> **Q6 (Minor issues):**  In Algorithm 1, the variable *list* is used but not introduced or defined.
>
> **A6:** Sorry for the ambiguity. The variables “list” and “Candidate” actually refer to the same variable. Thus, in the revised version of our paper (Line 221), we have replaced  “list” with “Candidate” to ensure consistency.
>
> Hope our detailed responses have properly addressed your concerns. We have incorporated your insightful comments into our revised version. If any concerns remain, please feel free to reach us at any time. We are happy to give further explanations.
>
> [1] Large language Diffusion Models, Nie et al, In NeurIPS 2025.

---

> ### Author Response · Authors · 2025-11-27
>
> Dear Reviewer GjSM,
>
> It is our great honor to have you as a reviewer for our paper. We sincerely appreciate the time and effort you have devoted to your review. We have carefully addressed all the weaknesses and concerns you raised—particularly the hyperparameter sensitivity, the derivation of Equations 7 and 8, theoretical implications, deeper analysis and evaluation on the SOTA frameworks.
>
> In our rebuttal and the revised manuscript, we have:
>
> - provided clear and practical guidelines for tuning all hyperparameters, together with extensive ablation studies across benchmarks.
> - rewritten the derivation of Equations (7) and (8) with more details and enhanced the intuitive explanation in both Sections 3 and 4.
> - added a new theoretical result (Theorem 1) to analyze the decoding error behavior of FDM and illustrate its advantage over heuristic baselines.
> - expanded the analysis in Line 239 to make the conclusion more interpretable and convincing.
> - further performed experiments with the lm-evaluation-harness framework to demonstrate that the benefits of FDM or FDM-A can be extended to SOTA evaluation frameworks.
> - fixed the minor issue on Algorithm 1 to ensure consistency.
>
> We sincerely hope that these revisions and clarifications have resolved all your concerns.
>
> As the discussion phase is nearing its end, we just wanted to gently check whether you have any remaining concerns. We would be very happy to clarify anything further if needed.
>
> Thank you again for your thoughtful feedback. Wishing you a lovely day!
>
> Warmly,
>
> Authors

---

### Author Response · Authors · 2025-11-23
**Summary of Paper Updates**

Dear Reviewers,

Thank you for your insightful and constructive feedback, which has helped us further improve the quality of our paper. Through thorough proofreading, we have corrected all bugs and mistakes in the submission. In addition, we have added new theoretical analysis and ablation studies to strengthen the contribution and practical value of our method:

- **Section 1 and 2**: We revised the wording and polished the presentation to improve clarity.
- **Section 3:** We corrected the issues in Equation 1 and added Equation 5, which serves as the basis for deriving Equations 9 and 10.
- **Section 4:** We introduced **a new theoretical result (Theorem 1)** showing that our proposed FDM achieves a lower divergence with the natural distribution compared with heuristic decoding. We also rewrote Lines 242–249 to make the conclusions more convincing and transparent, and further refined other imprecise expressions to improve clarity.
- **Section 5:** We add **new ablation studies** on $\eta_1$ and $\eta_2$ to analyze their effects on both accuracy and TPS.
- **(new) Appendix B:** We provide the complete proof of Theorem 1.
- **(new) Appendix E:** We include additional ablation studies on the HumanEval and ARC benchmarks to examine the influence of **$K$**, $\gamma$, $\eta_1$, $\eta_2$ on model performance.
- **(new) Appendix F:** We provide a theoretical analysis to explain the degradation in accuracy when $K$ is extremely large.

---

### Meta-Review · Area_Chair_WMWK · 2026-01-07

**Summary:**

The reviewers raised the several main concerns. First, there were serious issues with the paper's presentation, including numerous typos and mistakes, some of which appeared in the mathematical derivations. Second, reviewers questioned the practicality of the proposed approach due to the introduction of multiple new hyperparameters. Additionally, reviewers requested further detailed analysis and explanation of the method and experiments.

**Reviewer Concerns:**

While the authors have added guidelines for hyperparameter tuning of the proposed method, the need to tune 4 additional hyperaparameters likely remain a limitation of the current method. Moreover, the authors’ rebuttal resulted in substantial revisions, including significant changes to the technical sections and the addition of many new results. Given the extent of these changes relative to the original submission, the revised version effectively constitutes a new submission. As a result, the AC believes that a resubmission would be more appropriate to allow for a thorough and fair evaluation of the updated manuscript.

**Reviewer Scores:**

Given the substantial revisions introduced in the rebuttal, reviewers would likely require a full review of the revised manuscript in order to meaningfully update their scores.

---

### Decision · Program_Chairs · 2026-01-26

Reject